# The impacts of secondary ice production on the microphysics and dynamics of mid-latitude cold season convection

Zhipeng Qu[1], Alexei Korolev[1], Jason A. Milbrandt[1], Ivan Heckman[1], Mélissa Cholette[1], Cuong Nguyen[2], Mengistu Wolde[2]

[1]Meteorological Research Division, Environment and Climate Change Canada, Toronto, Ontario, Canada
[2]National Research Council Canada, Ottawa, Canada

*Correspondence to: Zhipeng Qu (zhipeng.qu@ec.gc.ca)*

**Abstract.** This study examines the impact of inclusion of secondary ice production (SIP) parameterizations on the cloud microphysics and dynamics in numerical weather prediction (NWP) simulations under mid-latitude winter conditions. Hindcast mesoscale model simulations were performed for two flights from the 2019 In-Cloud ICing and Large-drop Experiment (ICICLE) field campaign. The simulations used a horizontal grid spacing of 250 meters and employed a detailed triple-moment bulk microphysics scheme capable of predicting the liquid fraction of hydrometeors. SIP processes, including the Hallett-Mossop (HM), fragmentation of freezing drops (FFD) and fragmentation due to ice-ice collisional breakup (CB), are parameterized in this study. The NWP simulation results are compared with observational data collected during the ICICLE campaign. Sensitivity tests were conducted to highlight the importance of better quantifying SIP production rate in the NWP models. The findings indicate that SIP significantly enhances the simulated cloud ice number concentration and ice water content, particularly under strong convective conditions during winter. Additionally, the results reveal that the simulations are highly sensitive to the parameterization of HM, FFD and CB processes due to the interaction between these SIP mechanisms. High ice water content (HIWC) production is closely associated with SIP in strong convective conditions, whereas in stratiform conditions, HIWC can occur without a significant impact from SIP.

## 1 Introduction

Secondary ice production (SIP) is recognized as a fundamental process in cloud microphysics (e.g., Cantrell and Heymsfield, 2005; Field et al., 2017). SIP primarily enhances ice particle concentrations, often exceeding those from primary ice production (PIP) by several orders of magnitude (e.g., Hobbs and Rangno, 1985; Ladino et al., 2017). This significant increase in ice particle concentration can profoundly impact the phase composition, cloud dynamics, precipitation rates, and cloud radiative properties, thereby influencing the energy balance and hydrological cycle on both regional and global scales (Field et al. 2017, Kanji et al. 2017, Korolev and Leisner, 2020, Zhao and Liu, 2021). At present, seven mechanisms are recognized as sources of secondary ice in clouds. These include the fragmentation of freezing droplets (FFD) (e.g., Kleinheins et al., 2021), rime splintering (the Hallett-Mossop process, HM) (e.g., Hallett and Mossop, 1974), fragmentation due to ice-ice collisional breakup (CB) (e.g., Vardiman, 1978; Takahashi et al., 1995), ice fragmentation due to thermal shock (e.g., Dye and Hobbs, 1968),

fragmentation of sublimating ice (Oraltay and Hallett, 1989), activation of ice-nucleating particles (INPs) in transient supersaturation around freezing drops (e.g., Prabhakaran et al., 2020), and the break-up of freezing water drops upon impact with ice particles (James et al., 2021). The first six mechanisms are reviewed and summarized by Korolev and Leisner (2020). Among these, HM, FFD and CB are the most experimentally studied SIP mechanisms. However, detailed analyses of previous

experiments by Korolev and Leisner (2020) revealed a large variability in ice production rates, indicating that these SIP processes require further study. The other three mechanisms have been explored in a limited number of laboratory experiments, addressing only a subset of environmental conditions (e.g., fragmentation of sublimating ice) or merely demonstrating the general feasibility of SIP mechanisms (e.g., fragmentation due to thermal shock, activation of INPs in transient supersaturation around freezing drops). Consequently, Korolev and Leisner (2020) concluded that the relative contributions of each of the six

SIP mechanisms to the enhancement of ice concentrations remain uncertain.

In recent years, many studies have investigated the effect of SIP on cloud microphysics with the help of model simulations (e.g., Phillips et al., 2017a, 2017b, 2018; Sullivan et al., 2018; Hoarau et al., 2018; Fu et al., 2019; Sotiropoulou et al., 2020, 2021; Dedekind et al., 2021; Hawker et al., 2021; Huang et al., 2021, 2022; Qu et al. 2022; Lachapelle et al. 2024, 2025). Most of these modeling efforts were focused on matching simulated moments of particle size distributions (PSDs) with those

observed *in situ*. In many ways, the implementation of SIP in numerical models was hindered by the lack of consensus on parameterizations of SIP mechanisms.

Previous studies have also shown that the occurrence of high ice water content (HIWC) is linked to an increase in ice particle concentrations driven by SIP processes. HIWC conditions are defined as cloud environments with ice water content (IWC) exceeding 1 g m$^{-3}$. These conditions present significant hazards for civil aviation, potentially leading to engine power loss,

stalls, or structural damage (e.g., Lawson et al. 1998; Mason et al. 2006, Mason and Grzych, 2011). The phenomenon of HIWC is well documented from *in situ* observations in tropical mesoscale convective systems (MCSs) (e.g., Heymsfield and Palmer 1986, Lawson et al., 1998, Gayet et al., 2012, Fridlind et al., 2015; Leroy et al., 2017, Strapp et al., 2021). Several previous modeling studies using different cloud microphysical parameterizations attempted to reproduce high IWCs. Ackerman et al. (2015) used a 1D model to explore microphysics in tropical MCSs. Simulations performed with 3D models (Franklin et al.,

2016; Stanford et al., 2017; Qu et al., 2018) pointed to the inaccuracies in the estimation of cloud PSD, IWC, and ice category comparing to the observations. Huang et al. (2021) conducted high-resolution simulations of tropical convection and found significant overestimates of radar reflectivity (Z) and underestimates of total ice crystal concentration (N$_i$). Adding SIP in high-resolution simulations with advanced microphysical scheme, Huang et al. (2022) found significant improvement of the simulated Ni matching to the *in situ* observations. Similarly, Qu et al. (2022) conducted idealized simulation for tropical

convection and found that SIP significantly enhances the IWC and Ni, and reduces the simulated Z. The size of anvil cloud also increased when SIP is included in the simulation. Korolev et al. (2024), based on *in situ* observations and idealized high-resolution simulations, demonstrated that SIP in the vicinity of the melting layer plays a crucial role in the formation, maintenance, and longevity of HIWC in tropical MCSs. Recirculation of droplets through the melting layer by convective

updrafts was identified as a primary mechanism providing a high concentration of large, supercooled drops, which initiated a massive production of secondary ice due to the SIP FFD process.

Although most HIWC conditions occur in tropical regions, they can also be encountered in mid-latitude regions, even during winter, as HIWC-related aircraft issues have been reported. Rugg et al. (2022) examine these conditions using ICICLE observations and raise questions about the possible role of SIP in contributing to the formation of HIWC conditions. In addition to influencing HIWC conditions, SIP could also impact other cloud microphysical processes during winter. Cholette et al. (2024) assessed the impact of incorporating the HM process in 2.5 km resolution NWP simulations, demonstrating significant improvements in freezing rain forecasts. Similarly, Lachapelle et al. (2024) examined the influence of SIP on ice pellet simulations and reported enhanced predictive performance.

The first objective of this study is to examine the impact of inclusion of SIP in the simulation of mid-latitude winter convection, with a particular focus on the link between SIP and HIWC condition and how this compares to the impacts previously observed in tropical convection. Since different parameterization approaches for various SIP mechanisms can impact model simulation results, the second objective is to explore the sensitivity of parameterized mechanisms of HM, FFD and CB for mid-latitude winter conditions. This sensitivity study underscores the importance of better quantifying SIP production rates in nature under different atmospheric conditions.

The remainder of the paper is structured as follows. The next section describes the observational data used for evaluation. Section 3 details the model configuration, the microphysics scheme, and the parameterizations of SIP. Section 4 describes the various experiments conducted in this study. Section 5 evaluates the simulation results for two ICICLE flights. The differences of ice particle size distribution between mid-latitude winter convection and tropical convection are discussed in Section 6. The final section presents the conclusions and perspectives of the study.

## 2. Observation data

*In situ* data used in this study were collected from the National Research Council Canada (NRC) Convair-580 research aircrafts. The flight operations of the NRC Convair-580 in the frame of the ICICLE campaign were performed in the Great Lakes region of the United States from January to March 2019 (DiVito et al. 2020; Bernstein et al. 2021a, b). The ICICLE campaign focused on improving the understanding of aircraft icing hazards, specifically those caused by in-cloud supercooled large droplets (SLD).

The measurements of PSDs used in this study were performed by three particle probes, which covered different particle size ranges. The Droplet Measurement Technologies (DMT) Cloud Droplet Probe (CDP: Lance et al., 2010) was used for measurements of droplets in size range 2 $\mu$m < D < 50 $\mu$m. The Stratton Park Engineering Company (SPEC) 2D imaging-stereo (2D-S: Lawson et al. 2006) covered the nominal size range from 10 to 1250 $\mu$m. The DMT Precipitation Imaging Probe (PIP: Baumgardner et al., 2001) provided measurements of particle sizes and their concentrations in the nominal size range

from 100 $\mu m$ to 6.4 mm. The processing software employed a retrieval algorithm of partially viewed particle images (Heymsfield and Parrish, 1979; Korolev and Sussman, 2000), which allowed the enhancement of particle statistics and extended the maximum size of the composite PSD up to 12.8 mm.

All particle probes were equipped with anti-shattering tips to mitigate the effect of ice shattering on the measurements of ice particle concentration (Korolev et al., 2013). Residual shattering artifacts were identified and filtered out with the help of the inter-arrival time algorithm (Field et al., 2006; Korolev and Fields 2015). Cloud particle number concentration, characteristic sizes of ice and liquid particles, ice and liquid water content (LWC) were calculated from PSDs measured by particle probes. The sensitivity of cloud microphysical probes to cloud particle thermodynamic phase and cloud discrimination phase algorithm could be found in Korolev et al. (2017) and Rugg et al. (2022).

During the ICICLE project, the NRC Convair-580 was also equipped with the NRC Airborne W and X-bands (NAWX) radar system (Wolde & Pazmany, 2005). The radar reflectivity from the X-band of the NAWX radar is used to characterize the cloud macrophysical structure and assess the result of the SIP simulations.

## 3. Model simulation

### 3.1. Model configuration

The numerical model used in this study is the Global Environmental Multiscale (GEM) model (Côté et al., 1998; Girard et al., 2014). GEM is used for operational numerical weather prediction (NWP) in Environment and Climate Change Canada (ECCC) as well as research in ECCC and Canadian universities. The dynamical core of GEM is formulated based on the non-hydrostatic fully compressible primitive equations with a terrain-following hybrid vertical grid. As such, it can be run at cloud-resolving (sub-km grid spacing) scales. It can be run on global or limited-area domains and is capable of one-way nesting. In this study, four cascade domains with horizontal grid-spacings of 10, 2.5, 1.0 and 0.25 km and sizes of $3600\times2600$ km$^2$, $1800\times1800$ km$^2$, $1024\times1024$ km$^2$ and $512\times512$ km$^2$ are used respectively. The high-resolution simulation with a grid spacing of 0.25 km provides enhanced capability to resolve small-scale phenomena such as convective updrafts and the recirculation of raindrops by these updrafts near the melting layer. Accurately resolving these physical processes is crucial for better representing SIP mechanisms, including HM, FFD and CB processes (Korolev et al., 2024). For ICICLE Flight 9, the simulation is initiated at 12:00 UTC on 2019-02-07 and runs for 12 hours. All four nesting domains are centred on 44.5ºN and 88.75ºW (Figure 1). For ICICLE Flight 20, the simulation is initiated at 06:00 UTC on 2019-02-23 and runs for 12 hours, with all four nesting domains centred on 42.7°N and 89.9°W (Figure 1). 84 unevenly spaced hybrid vertical levels are used for 10 km simulations and 62 levels are used for all other simulations.

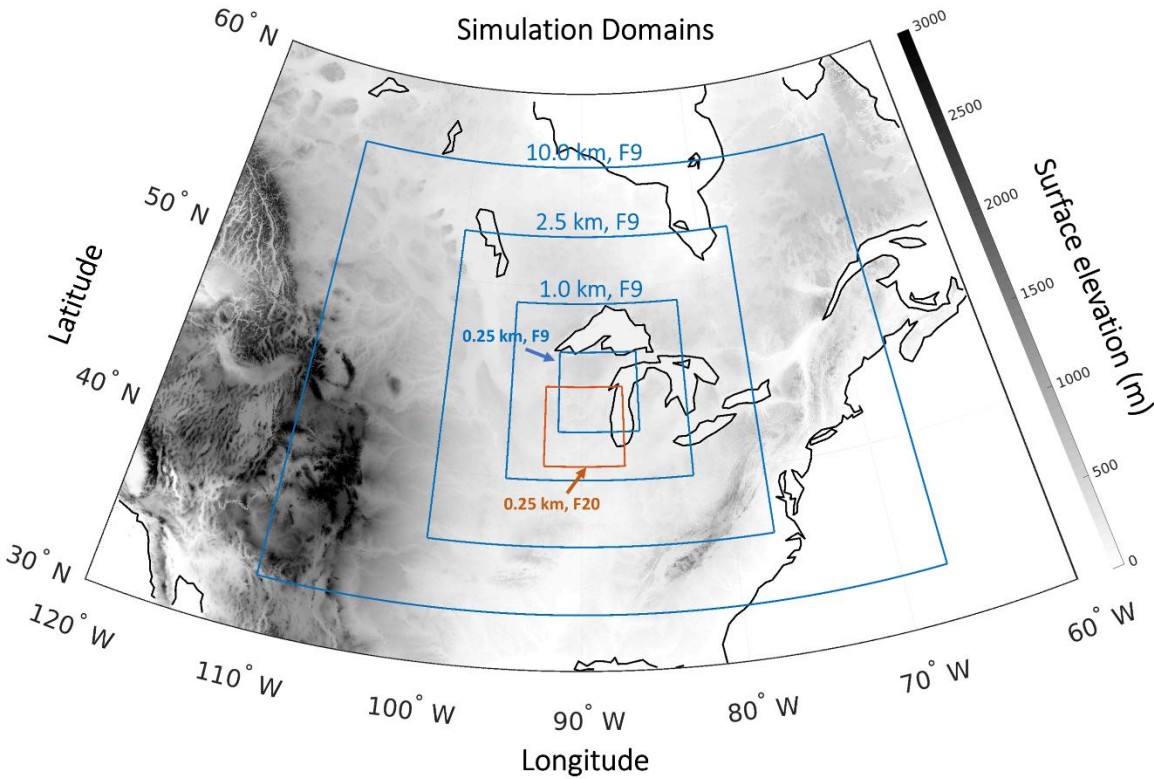

Figure 1. Simulation domains at different stages of the cascade for ICICLE Flight 09 (blue rectangles), ranging from 10 km to 0.25 km resolution. The red rectangle marks the innermost 0.25 km domain for Flight 20. The three outer domains for Flight 20 follow a similar pattern to Flight 09 but are shifted, as they center on the red rectangle. The grayscale shading represents surface elevation in meters, while black lines indicate waterfront contours.

## 3.2. Cloud microphysics scheme

In the GEM simulations of this study, all grid-scale cloud microphysical processes were represented by the Predicted Particle Properties (P3) bulk microphysics parameterization scheme. The original version of the P3 scheme is described in detail in Morrison and Milbrandt (2015). Since its inception, there have been several advances which have been described in various publications.

The liquid-phase component of P3 uses a standard two-category, two-moment approach whereby all liquid-phase hydrometeors are partitioned into "cloud" (non-sedimenting droplets) and "rain", each of whose PSDs are represented by complete gamma functions and each has the mass and number mixing ratios as prognostic (and advected) variables. Droplet activation is parameterized following the approach of Morrison and Grabowski (2007, 2008), which assumes a constant background aerosol concentration of 300 $cm^{-3}$ with a mean particle size of 0.05 micrometers (μm), composed of ammonium sulfate. Work is currently underway to include prognostic aerosols.

P3 has a unique approach to represent ice-phase hydrometeors. In the traditional approach, as used in nearly all bulk and bin microphysics schemes, particles are partitioned into predefined categories (e.g. "ice", "snow", "graupel", etc.) with prescribed physical properties (e.g. density, fall speed parameters, etc.). In contrast, P3 has a user-specified number of freely evolving generic ice-phase categories, each of whose physical properties evolve continuously in time and space. Because of this, each ice category in P3 can represent any type of ice-phase hydrometeor, although, as in any bulk scheme, ice categories in P3 remain constrained by assumed PSD parameters type and the number of prognostic moments. Also, there is no artificial "conversion" between ice-phase categories, which is an inherent weakness in traditional schemes. Similar to the liquid categories, the PSD of each ice category in P3 is represented by a complete gamma function but has an optional triple-moment approach (Milbrandt et al. 2021). The generalization to the multi-category version of P3 is described in Milbrandt and Morrison (2016). The incorporation of an optional prognostic liquid fraction, which allows for mixed-phase particles and improves the simulation of processes such as melting and refreezing, is described in Cholette et al. (2019, 2024).

All simulations in this study use the configuration with triple-moment ice, prognostic liquid fraction, and four ice-phase categories. The decision to use four ice categories was made to maximize the representation of the multi-modal ice particle size distribution during conditions of high SIP rate where large pre-existing ice particles coexist with small ice splinters. Ice splinters generated by various SIP processes are assumed to have a diameter of 10 μm. These small ice particles are incorporated into one of the four ice categories corresponding to the closest particle size. Qu et al. (2022) showed that three ice categories in the P3 scheme are sufficient to effectively capture the size distribution when SIP is active; Milbrandt et al. (2025) showed a similar "convergence" with three categories for the simulation of hail. Nevertheless, four categories were used in this study since the additional category should, in principle, model the total ice phase slightly better without significantly increasing computational cost.

In the "official" P3 scheme, ice is initiated via homogeneous freezing of cloud droplets, immersion/contact freezing of cloud droplets and rain drops (temperature-dependent following Bigg (1953) with parameters from Barklie and Gokhale (1959) ), heterogeneous nucleation through condensation freezing of cloud droplets and rain drops, and deposition nucleation (Cooper, 1986), and rime splintering (Hallet and Mossop, 1974). Deposition nucleation is specifically restricted to temperatures at or below 258.15 K and requires ice supersaturation of at least 5%.

### 3.3. Parameterization of SIP

This study focuses on assessing the impact of including three SIP processes in the NWP simulations: the HM rime splintering process, the FFD process and the CB process. As discussed, additional SIP processes are currently known; however, due to limited understanding of each, including HM, FFD and CB, this study examines only these relatively better-characterized processes. Future studies will aim to incorporate more SIP processes.

### 3.3.1 Rime splintering/Hallet-Mossop (HM)

The parameterized HM process follows Hallet and Mossop (1974) stating the production of a peak value of 350 ice splinter per mg of collected liquid water during riming of liquid cloud drops or rains droplets within a temperature range of -3°C > T > -8°C, with the peak value at -5°C and varying linearly to 0 at -3 and -8 °C. The size of ice splinters is set to 10 μm. This study utilizes three distinct methods to calculate the number of ice splinters produced by the HM process. In the first method (referred to as HMr), the number of ice splinters is determined based on the mass of all collected supercooled raindrops,

excluding the fraction involved in the FFD process (section 3.3.2). The second method (referred to as HMgr) calculates the splinters using the mass of raindrops collected by graupel, while the third method (referred to as HMgc) uses the mass of liquid cloud droplets collected by graupel. In the P3 scheme, ice is represented as generic categories, so there is no predefined graupel category; therefore graupel is identified as ice (in a given category) with a rime fraction greater than 0.4, a bulk ice density between 50 and 700 kg m⁻³, and a mean-mass diameter greater than 2 mm. Liquid "cloud droplets" are defined as having a

diameter smaller than 50 μm, while "rain drops" have a diameter exceeding this threshold.

### 3.3.2 Fragmentation of Freezing Drops (FFD) process

The FFD process is parametrized following Lawson et al. (2015):

$$N_f = 2.5 \times 10^{-11} D^4, \tag{3}$$

where $N_f$ is the statistical average number of ice fragments per drop (#/freezing drop), and $D$ is the freezing liquid drop diameter

in μm. Two versions of FFD parameterization were tested in this study. In the first version, the FFD process was applied for raindrops (100 μm < D < 3500 μm) which were nucleated by contacting with ice particles with diameter $D_i$ < 100 μm. Because freezing raindrops up to 3500 μm in diameter were included in the calculation of ice splinters, and the FFD rate is proportional to $D^4$, this approach can yield very high FFD rates. This version is therefore referred to as FFDh (FFD at a higher rate).

In the second version, a correction to Eq. 3, proposed by Lachapelle et al. (2025), was implemented. It was found that the

190 original FFD rate in Eq. 3 could be overestimated compared to in situ observations from the Winter Precipitation Type Research Multiscale Experiment (WINTRE-MIX) campaign during ice pellet precipitation (Minder et al. 2023). To address this, the value of $N_f$ was capped for raindrop diameters equal to or greater than 1878.2 μm. This adjusted version is referred to as FFDl (FFD at a lower rate) in this study.

Following Keinert et al. (2020) the activity of the FFD process was limited to the temperature range -25°C < T < -2°C with a

195 maximal rate of $N_f$ (as calculated by Eq. 3, with or without capping) at T = -12.5°C. $N_f$ is then linearly scaled down to zero as the temperature decreases from −12.5°C to −25°C or increases from −12.5°C to −2°C, according to a T dependent scaling factor c (where c∈[0,1]), such that the rate is c×$N_f$. All newly produced ice splinters have a diameter of 10 μm.

A minimum ice mixing ratio threshold of $5\times10^{-5}$ kg kg$^{-1}$ was applied to activate the FFD process. The choice of this threshold was determined subjectively by trial-and-error. Its application prevents the unintended initiation of a high FFD production rate under freezing rain conditions due to the presence of very small amounts of ice. In freezing rain conditions, where large amounts of supercooled liquid raindrops are present, even a small amount of ice could trigger a chain reaction, leading to an unrealistically high ice number concentration.

In the FFD parameterizations used here, supercooled rain collected by ice will freeze while producing ice splinters. Excluding the mass of ice splinters, which typically represents a small fraction of the rain mass, the remaining rain mass is added to the ice category that collected the rain. Under extreme conditions, such as when a small number of large raindrops mix with a large number of tiny ice particles, the collected rain mass added to the category of ice with very small mean-mass diameter which collect rain droplets can result in the production of a large number of mid-sized ice particles. In reality, this scenario would produce a few large ice particles and many small ones. To mitigate this effect, and following the methodology proposed by Lachapelle et al. (2024), we redirect the mass of collected rain with large mean mass-weighted diameter to the ice category that is most similar in size. This redirection takes place when the mean mass-weighted diameter of the rain is more than twice that of the ice that collect them.

In conditions with large raindrops, the FFD parameterizations used in this study may produce an excessively high number of ice splinters. To limit these unrealistically large values of ice number mixing ratios, a maximum total threshold of $5 \times 10^7$ kg$^{-1}$ (summed across all four ice categories) is therefore applied in all simulations.

### 3.3.3 Fragmentation due to ice-ice collisional breakup (CB) process

The CB process was parameterized following Phillips et al. (2017b). It is governed primarily by the collision kinetic energy (CKE) and is also influenced by air temperature as well as the type and size of the colliding ice particles. All three types of CB processes were implemented: type I (collisions between graupel/hail and other graupel/hail), type II (collisions between crystals/snow and graupel/hail), and type III (collisions between crystals/snow and other crystals/snow). All newly produced ice splinters were assumed to have a diameter of 10 µm. An additional modification was made to ensure that CB-generated ice splinters were only calculated when the relative velocity between the colliding ice particles exceeded 1 m s$^{-1}$ (Hoarau et al. 2018, Korolev and Leisner, 2020).

### 4. Establishment of experiments

An objective of this study is to examine the impacts of different approaches to modeling the HM, FFD and CB processes. The baseline simulation (referred to as "BASE") is conducted with all SIP processes deactivated. Four additional simulations were carried out: three including both the HM and FFD, and one including HM, FFD and CB. In the three two-process experiments, the same FFDh (higher rate) parameterization is used, coupled with three different HM parameterizations as described in

Section 3.3.1. In the final experiment, all three SIP processes were enabled, with the FFDl (lower rate) parameterization applied. A complete summary of all configurations is provided in Table 1. The 0.25 km simulations are designated as "[case]-[configuration]," where the case refers to either "F9" or "F20," representing ICICLE Flight 9 or Flight 20, respectively. The configurations correspond to the names listed in the left column of Table 1. For instance, 'F9-BASE' refers to the 0.25 km baseline simulation for ICICLE Flight 9.

**Table 1. List of simulations.**

| Experiment name | HM | FFD | CB |
|---|---|---|---|
| BASE | off | off | |
| HMr-FFDh | HM (based on ice collected rain drops, excluding collected rain used for FFD), temperature dependent | Temperature & liquid drop size dependent. The original formula of Lawson et al. (2015) was used (higher rate). | off |
| HMgr-FFDh | HM (based on graupel collected rain drops), temperature dependent | | |
| HMgc-FFDh | HM (based on graupel collected liquid droplets),temperature dependent | | |
| HMgc-FFDl-CB | | Modified version of Lawson et al. (2015) based on Lachapelle et al. (2025) (lower rate) | Based on Phillips et al. (2017b) |

## 5. Results

This section begins by presenting the results of ICICLE Flight 9, characterized by localised convections with peak updraft speed exceeding 5 m s⁻¹. It then examines ICICLE Flight 20, which exemplifies a stratiform mesoscale system, with vertical wind velocities below 5 m s⁻¹.

### 5.1. ICICLE Flight 09: localized convective case

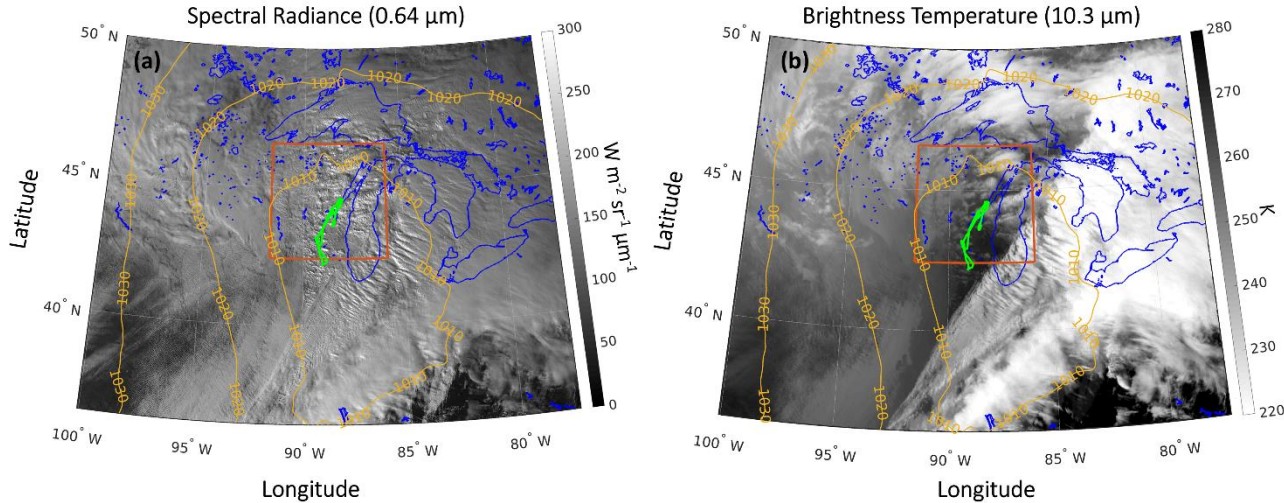

Figure 2. (a) GOES-16 spectral radiance at 0.64 μm (Channel 2). (b) GOES-16 brightness temperature at 10.3 μm (Channel 13). Both images are from 20:30 UTC on 9 February 2019. Blue lines denote coastlines and lakes; the red rectangle indicates the inner-most simulation domain (with a 0.25 km grid spacing) for Flight 9; yellow lines show the sea level pressure isobar at 18:00 UTC from the operational regional analysis used at ECCC; green lines: aircraft flight track. (Only data from the flight
segments in the northern portion of the track were used in this study).

On 9 February 2019, the Great Lakes region experienced a significant winter storm fuelled by a low-pressure system that intensified as it stalled over Lake Huron (Figures 2a and 2b). This system resulted in prolonged snowfall across the region. In the area west of Lake Michigan, including Wisconsin and northern Illinois, cold temperatures and snowy conditions prevailed due to the influence of a passing low-pressure system. A comparison between the visible radiances (Figure 2a) and infrared
brightness temperatures (Figure 2b) from GOES-16 satellite imagery reveals scattered convective cells. These isolated convective updrafts, though sparse, were capable of producing brief bursts of snow or freezing rain.

Figure 3 shows the combined zenith and nadir pointing X-band radar reflectivity along with *in situ* observations of IWC, ice number concentration ($N_i$, excluding particles smaller than 40 μm in diameter), LWC, air temperature (T), and the aircraft's altitude (H). The bright band at approximately 2.5 km is clearly visible before 20:00 UTC, but it disappears after 20:30 UTC.
A temperature inversion is present near an altitude of 2 km, although the maximum temperature at this altitude remains below freezing after 20:00 UTC. Based on *in situ* observations there are many regions of mixed-phase clouds, particularly between 19:50 and 20:40 UTC and around 21:00 UTC, which provide potential favourable conditions for SIP.

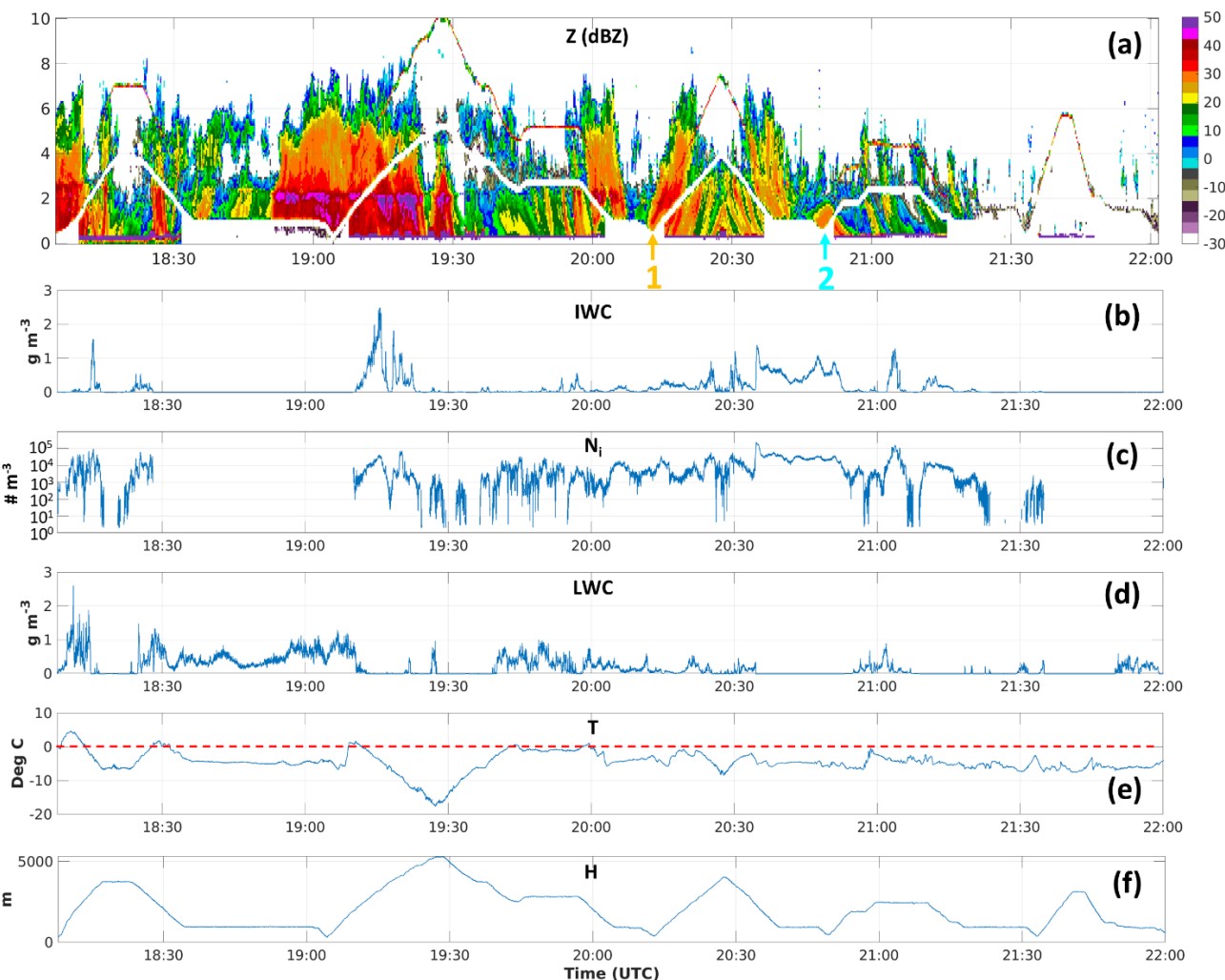

Figure 3. Data of ICICLE Flight 9 on 2019-02-07. (a): upward and downward X-band radar reflectivity (Z), (b): IWC, (c): $N_i$, (d): LWC, (e): T, with red dashed line indicating T=0°C, (f) H. The yellow and turquois arrows point to the two moments when Convair-580 flew a low altitude below 1 km during missed approach.

Figure 4 displays the simulated brightness temperature at 10.3 μm for the 2.5 km resolution domain (northern border limited at 50° N due to GOES-16 data availability) using the RTTOV (Radiative Transfer for TOVS) fast radiative transfer model (Saunders et al., 2018) for the F9-BASE simulation (Figure 4a). This is compared with the GOES-16 observation at 20:30 UTC on 9 February 2019 (Figure 4b). The F9-BASE simulation effectively reproduces the large-scale sea-level pressure pattern (shown by yellow lines). The simulated brightness temperature at 10.3 μm closely matches the GOES-16 observations. However, a positive bias (indicating warmer values) is evident compared to the satellite data. Investigations into the cause of

this bias are currently underway. Additionally, the simulation slightly underestimates cloud coverage over the southern part of Lake Michigan and in the southeastern corner of the simulation domain.

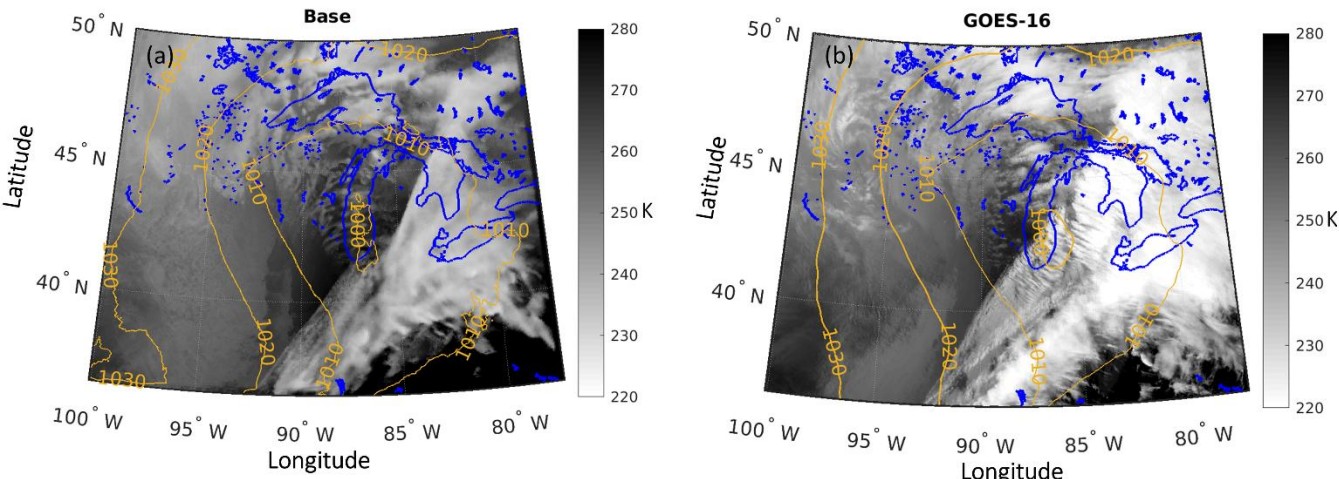

Figure 4. (a) Brightness temperature at 10.3 μm simulated using RTTOV (Saunders et al., 2018) with the BASE simulation at 2.5 km resolution for 20:30 UTC on 9 February 2019. (b) Observed brightness temperature from GOES-16 for 20:30 UTC on 9 February 2019. Blue lines indicate lake boundaries, while yellow lines represent sea-level isobars: from the F9-BASE simulation at 20:30 UTC in panel (a) and from ECCC's regional prediction model at 20:30 UTC initiated at 18:00 UTC in panel (b).

Figures 5 and 6 present the simulation profiles for the region within 10 km on each side of the aircraft track. The profiles display averaged values, except for Figure 5c, which shows 99 percentile of ice water content, the maximum vertical wind speeds (Figure 5d), and the median radar reflectivity ($Z_{median}$, Figure 6c). Significant variations in mean IWC among different simulations can be seen in Figure 5a. The F9-BASE simulation for ICICLE Flight 9 produces the lowest IWC, with slightly higher values observed in F9-HMgr-FFDh. In contrast, F9-HMgc-FFDh and F9-HMgc-FFDl-CB exhibit notable increases in IWC.The F9-HMr-FFDh simulation exhibits extremely high IWC near the surface. All SIP simulations produced a significant enhancement in the 99th percentiles of IWC at various altitudes compared to the F9-BASE simulation. The extremely high IWC observed below 1.8 km in the F9-HMr-FFDh simulation (below the temperature inversion at ~2 km altitude, as shown in Figure 5e) aligns with the elevated $N_i$ in the same altitude range (purple dash-dotted lines in Figure 5c). This IWC enhancement is strongly associated with increased freezing efficiency of liquid water (LWC and rain water content - RWC), as shown in Figure 6a and 6b, and enhanced vapor deposition growth due to the higher $N_i$, leading to a larger ice surface area for deposition. These findings suggest that SIP processes may contribute to the formation and persistence of HIWC condition.

For $Z_{median}$ (Figure 6c), the F9-BASE and F9-HMgr-FFDh simulations produce the highest values, despite having the lowest IWC. Due to the relatively low $N_i$ in these simulations, the ice particles in F9-BASE and F9-HMgr-FFD are larger compared to those in the other three SIP simulations. F9-HMgc-FFDh and F9-HMgc-FFDl-CB generate the lowest $Z_{median}$ across most altitudes below 5 km, while F9-HMr-FFDh shows even lower $Z_{median}$ below approximately 0.9 km. This reduction in $Z_{median}$ is

due to the extremely high $N_i$ in F9-HMr-FFDh, which significantly reduces the size of the ice particles. By 20:30 UTC, the clouds near the flight track had already glaciated, so no bright band could be seen in Figure 6c.

Although all three two-process SIP simulations use the same FFDh parameterization, differences in how the HM process is modeled result in markedly different outcomes. Figures 6d and 6e illustrate the HM and FFD rates at the simulation time step of 20:30 UTC. At this time, there is no SIP activity within 10 km of the flight track for F9-HMgr-FFDh. Similar to F9-HMgr-FFDh, F9-HMr-FFDh calculates the number of ice splinters based on collected rain mass, but without the requirement that the ice must be graupel. By omitting this graupel condition, F9-HMr-FFDh produces a high HM rate (Figure 6d, purple dash-
dotted line), which subsequently triggers the FFD process, leading to an even higher FFD rate near 1 km altitude. On the other hand, F9-HMgc-FFDh generates a lower HM rate near both 3.5 km and 1 km altitudes (Figure 6d, red lines). Consequently, the FFD rate is also very low near 3.8 km, with no FFD activity below 2 km near the flight track at this time step. These results indicate that under freezing conditions with an abundant supply of supercooled rain, the initiation of the FFD process is highly dependent on the rate of the HM process, or potentially any SIP processes which could provide sufficient amount of small ice
particles.

Including the CB process in the experiment led to an explosive increase in FFD rates near the surface (not shown). To mitigate this, we applied the FFDl (lower rate) parameterization, as proposed by Lachapelle et al. (2025). Results from the three-process experiment (F9-HMgc-FFDl-CB) were similar to those of F9-HMgc-FFDh in terms of $N_i$, IWC, LWC, and $Z_{median}$. The CB process (types I, II, and III) produced positive SIP rates at multiple altitudes, including levels below 2 km (Figure 6f, 6g), potentially further activating the FFD process. The use of the FFDl parameterization effectively suppressed near-surface FFD
rates, resulting in values that were substantially lower and more realistic.In the simulations for Flight 9, condensation freezing/deposition ice nucleation likely initiated ice particle formation at altitudes above 4.5 km (T<-15°C, as implemented in P3 scheme). As these ice crystals descend, they tend to aggregate, leading to decreased $N_i$ values (typically less than $10^2$ m$^{-3}$) (see Figure 7b, blue dashed line from the F9-BASE simulation). In addition, the immersion/contact freezing process could
also introduce low number of $N_i$ at higher T (T<-4°C). Ice particles produced via heterogeneous nucleation can subsequently trigger the HM or CB processes. However, since $N_i$ generated by heterogeneous nucleation remains low at lower altitudes, where supercooled liquid drops are abundant, the likelihood of activating significant FFD rates is limited. In contrast, the HM and CB processes can produce $N_i$ values at least an order of magnitude higher than those from heterogeneous nucleation alone, substantially increasing the potential for interactions between ice splinters and supercooled raindrops.

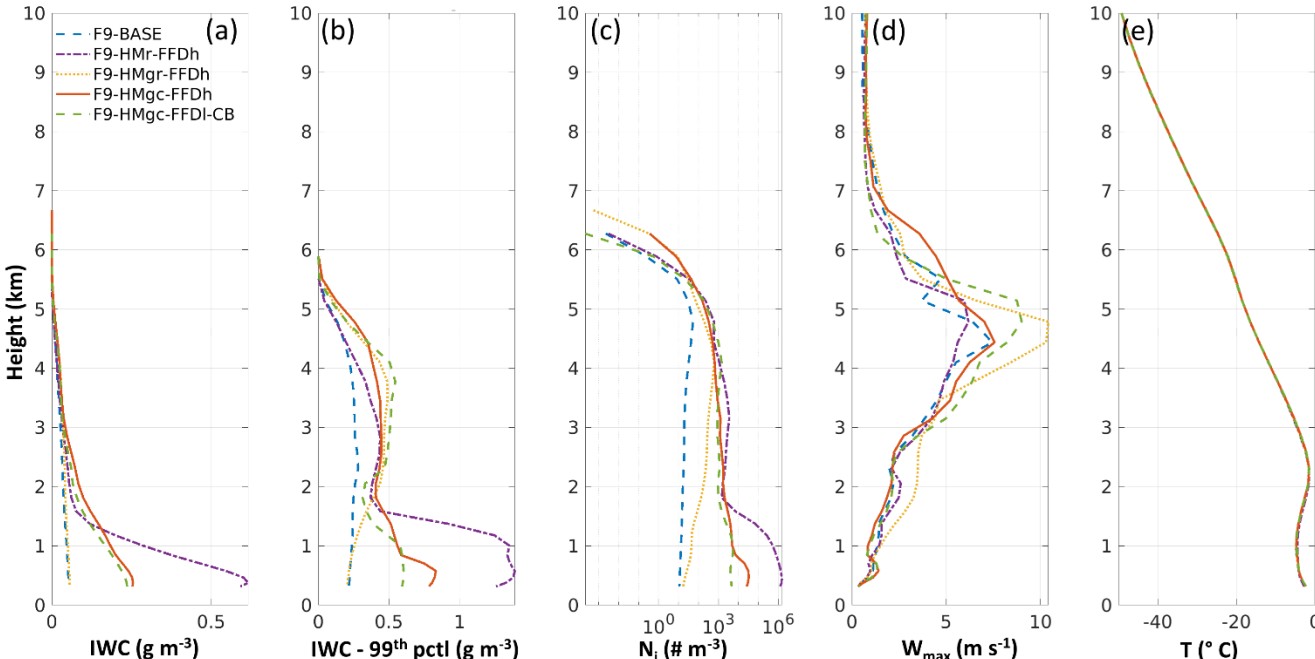

Figure 5. Profiles from the baseline and SIP simulations on 2019-02-07 for the case of ICICLE Flight 9. (a): mean IWC, (b): 95 percentile of IWC ($99^{th}$ pctl of IWC), (c): mean $N_i$ with the number of ice particle smaller than 40 μm excluded, (d): maximum vertical wind speed ($w_{max}$), (d) T (the three simulations have only slightly different temperature that is not distinguishable in the figure). All profiles are calculated from simulation results at 510 min (20:30 UTC) after the initiation time of 12:00 UTC. All profiles are calculated for the region within 10 km of distance to the aircraft track.

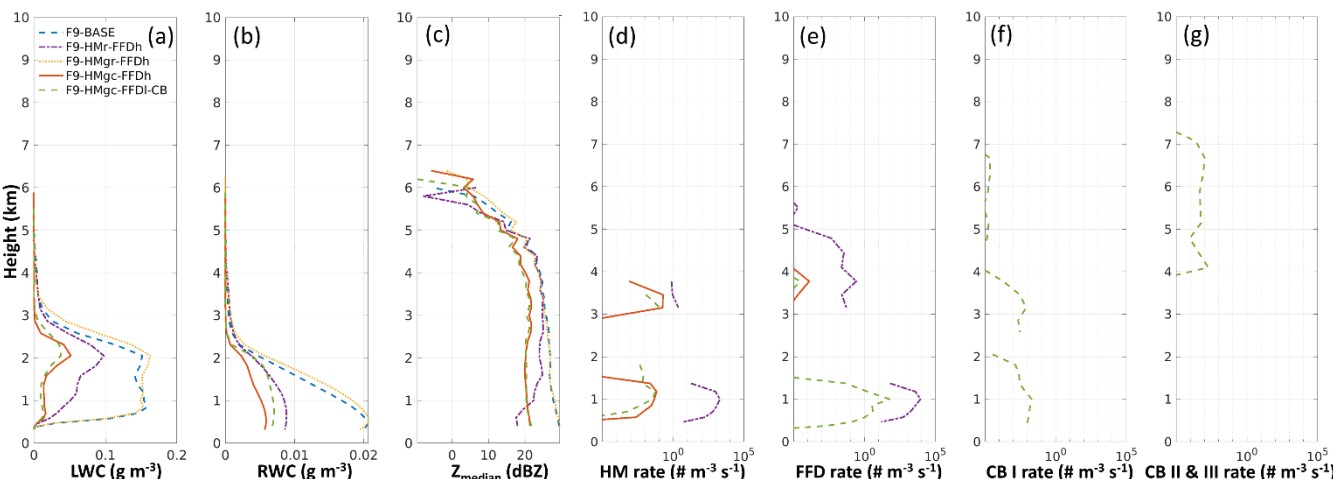

Figure 6. Profiles from the baseline and SIP simulations on 2019-02-07. (a): mean LWC, (b): mean RWC, (c): median radar reflectivity ($Z_{median}$), (d): mean Hallett-Mossop rate, (e): mean Fragmentation of Freezing Drop rate, (f): ice-ice collisional

breakup rate (type I), (g): ice-ice collisional breakup rate (type II & III) All profiles are calculated for the region within 10 km of distance to the aircraft track and at 20:30 UTC.

To further assess the simulation results, *in situ* aircraft data were used for comparisons. Figure 7 compares the distributions of simulated IWC and $N_i$ within 10 km of the flight track and within the flight altitude range of 0.5 to 3.8 km, with the observations shown by black lines. In the simulation results presented here, quantities from all four ice categories are summed and for the simulated $N_i$, ice particles smaller than 40 μm are excluded.

For IWC, the F9-BASE simulation significantly underestimates the frequency of clouds with IWC > 0.1 g m$^{-3}$. The F9-HMgr-FFDh simulation (yellow dotted line) shows some improvement, especially for IWC > 0.5 g m$^{-3}$, though most frequencies remain underestimated. While both F9-HMgr-FFDh and F9-HMgr-FFDl-CB considerably improve the estimated IWC frequency, they still underestimate conditions of high IWC. Conversely, F9-HMr-FFDh overestimates the frequency of high IWC above 0.8 g m$^{-3}$.

For $N_i$, both the F9-BASE and F9-HMgr-FFDh simulations produce very low values, while F9-HMr-FFDh (purple line) significantly overestimates $N_i$. The F9-HMgc-FFDh and F9-HMgr-FFDl-CB simulations (red line and green dashed line respectively) provide the closest estimates of $N_i$, although the peak values remain slightly lower than observed. Based on this, the subsequent discussion on SIP simulations focuses exclusively on F9-HMgc-FFDh and F9-HMgc-FFDl.

Figure 8 presents the two-dimensional histograms of $N_i$ (x-axis) as a function of T (y-axis). In the F9-BASE simulation (Figure 8a), the majority of $N_i$ values are low, ranging between $10^1$ and $10^2$ m$^{-3}$, predominantly at temperatures above -10°C. Because SIP processes are not included in this simulation and the cloud top remains warmer than -40°C (the threshold for homogeneous nucleation), ice particles primarily originate from heterogeneous ice nucleation. In the P3 scheme, the parameterization by Cooper et al. (1986) is applied for temperatures below –15°C and ice supersaturation greater than 5% to estimate $N_i$ due to condensation freezing and deposition ice nucleation. Since this scheme tends to overestimate $N_i$ at lower temperatures, $N_i$ values due to condensation freezing/deposition ice nucleation are capped at $1 \times 10^5$ m$^{-3}$. Therefore, one of the sources of ice particles observed at temperatures warmer than -10°C might have originally formed at colder temperatures below -15°C, governed by the estimated $N_i$ which is represented by the line of small red circles. As these primary ice particles descend, they aggregate, resulting in the low $N_i$ values at temperatures warmer than -10°C. Another source of ice could be from the immersion/contact freezing of cloud droplets or rain drops (T<-4°C).

The two simulations that include SIP processes, F9-HMgc-FFDh and F9-HMgc-FFDl (Figures 8b and 8c respectively), still exhibit similar cluster (highlighted by large red circles) as in the F9-BASE run, representing ice particles resulting from heterogeneous nucleation. Additionally, a second distinct cluster appears (highlighted by large green circles in Figure 8b and 8c), displaying higher $N_i$ values at slightly warmer temperatures (i.e., lower altitudes). This pattern contrasts with the expected reduction of $N_i$ at lower altitude due to aggregation, suggesting a strong influence of SIP on enhancing $N_i$.

Finally, the observed distribution shown in Figure 8d is very similar to those in the two SIP simulations, but with even higher $N_i$ values near $T = -5°C$ (between $10^3$ to $10^5$ m$^{-3}$). This pattern further underscores the pronounced impact of SIP on increasing ice particle concentrations.

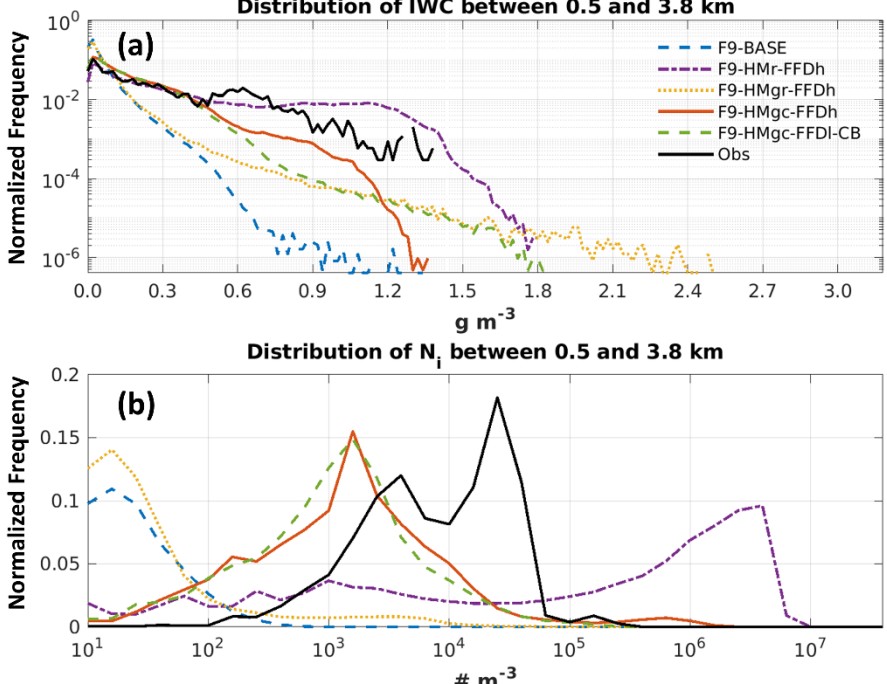

Figure 7. Distribution of IWC (a) and $N_i$ with the number of ice particle smaller than 40 μm excluded (b) for model simulations at 20:30 UTC on 2019-02-07 and for the observation data for ICICLE Flight 9 between 20:00 and 21:00 UTC. The results from model simulations are calculated for altitudes between 0.5 and 5.5 km and with ice water content higher than 0.001 g m$^{-3}$ in the region within 20 km of distance to the aircraft track. The results from *in situ* observation are calculated for the condition with IWC higher than 0.001 g m$^{-3}$. (b): logarithmic bin width of 1/5 of an order of magnitude is used. Blue dashed lines: Baseline simulation, purple dash-dot line: SIP (F9-HMr-FFDh), yellow dotted lines: SIP (F9-HMgr-FFDh) simulation, red lines: SIP (F9-HMgc-FFDh) simulation, green dashed lines: SIP (F9-HMgc-FFDl-CB), black lines: observation.

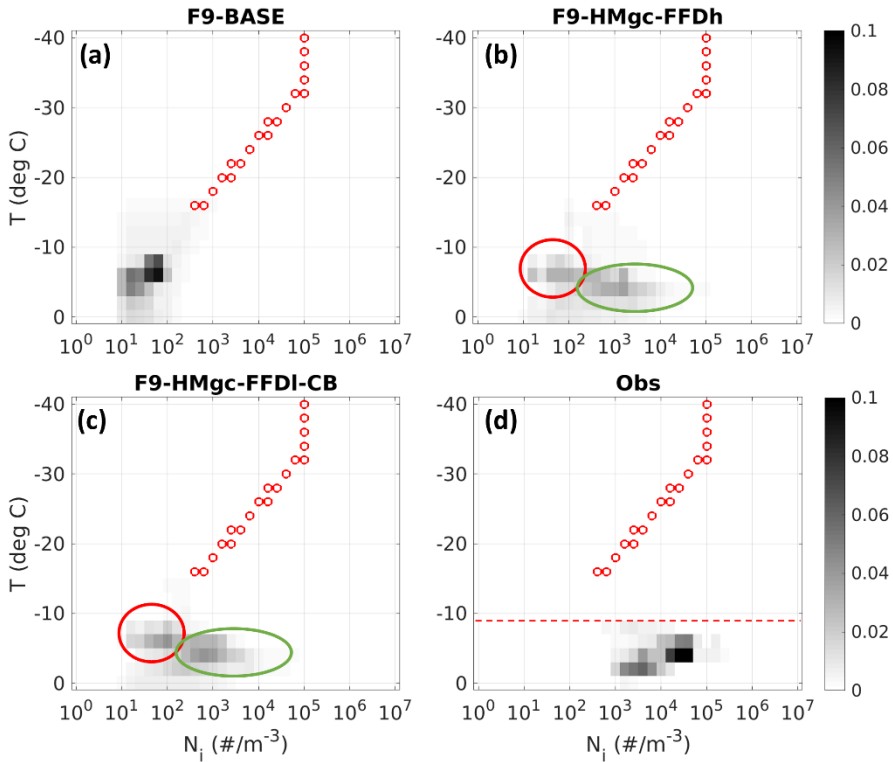

Figure 8: 2D histograms for the frequency of $N_i$ regarding the T for F9-BASE (panel a), F9-HMgc-FFDh (panel b), F9-HMgc-FFDl-CB (panel c), and the observation for Flight 9 between 20:10 and 20:50 UTC (panel d). Small red cycles: parameterized $N_i$ due to condensation freezing/deposition ice nucleation (adapted from Cooper, 1986). Red dashed line: minimal flight T between 20:10 and 20:50 UTC.

Comparing the simulated and observed Z values shows that both F9-HMgc-FFDh and F9-HMgr-FFDl-CB simulations resulted in a closer match to the observation data (Figure 9). The F9-BASE simulation produces $Z_{median}$ values 5 to 10 dBZ higher than observed below 5 km. In contrast, F9-HMgc-FFDh and F9-HMgr-FFDl-CB show slight underestimations below 2.5 km and small overestimations between 2.5 and 5 km. There may be artifacts related to ground clutters which cause enhanced Z at altitudes twice the aircraft height, as indicated in Figure 3a. The values of $Z_{median}$ shown in Figure 9e suggest that the simulations
might underestimate above 5 km. This underestimation will be further discussed in section 5.2.

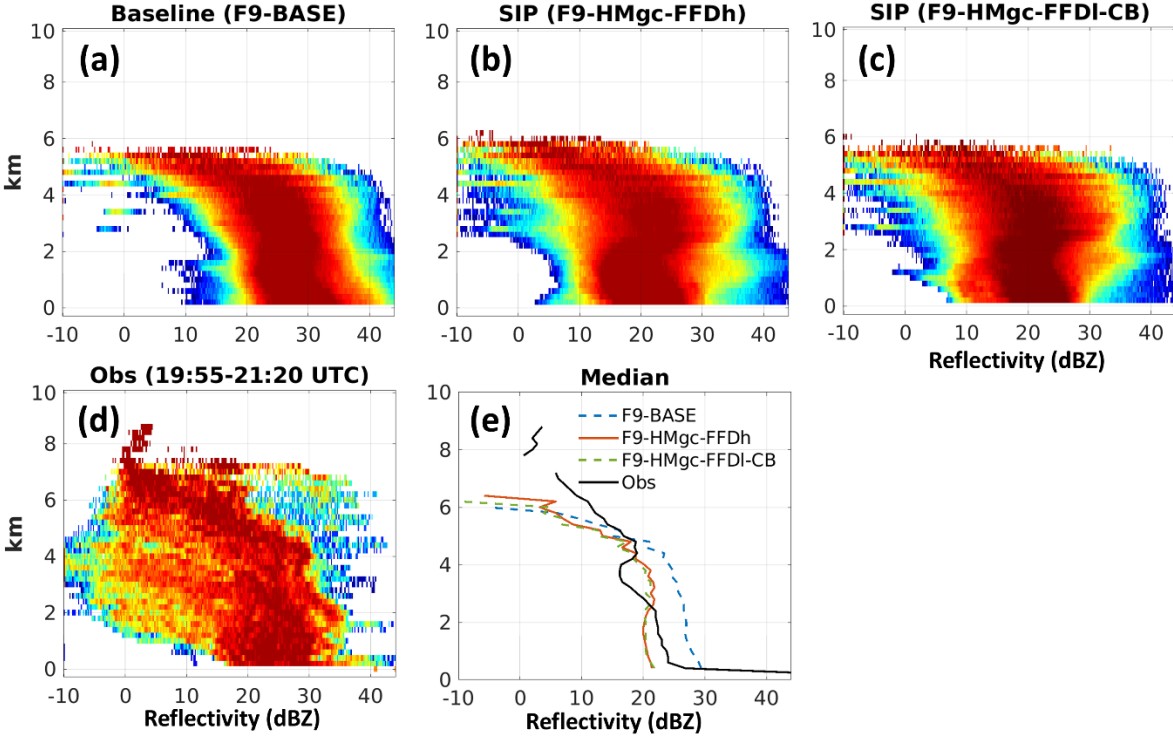

Figure 9. Z distribution frequency for (a): Baseline simulation, (b): SIP (F9-HMgc-FFDh) simulation, (c): SIP (F9-HMgc-FFDl-CB),(d): observation from Flight 9 between 20:10 and 20:50 UTC, and (e): the median values for each altitude for the three model simulations and observations.

Figure 10 shows the averaged values of IWC and $N_i$ for the F9-BASE (panel a, d), F9-HMgc-FFDh (b, e) and F9-HMgr-FFDl-CB (c, f) simulations between altitudes of 0.5 and 3.8 km. Comparing Figure 10a and 10b/c, there are noticeable local enhancements of IWC, particularly in the western part of the simulation domain (indicated by the white circles), where larger amounts of LWC and RWC are available (not shown). A similar pattern is observed for $N_i$ (Figure 10d, e, f). Despite this being a winter case, some of these updrafts reach speeds up to approximately 7 m s$^{-1}$, as shown in Figure 5d. Thus, convective updrafts may play a crucial role in creating favourable conditions for SIP even under winter conditions.

Active SIP processes not only enhance IWC and $N_i$ but also accelerate the freezing of existing liquid water, as shown in Figure 6a and 6b. This can potentially lead to a significant reduction in the area of freezing rain near the surface. Figure 11 presents the RWC and IWC near the surface at the lowest model layer (approximately 20 m above surface) for the F9-BASE (first column), F9-HMgc-FFDh (middle column) and F9-HMgr-FFDl-CB (third column) simulations. The black contours indicate areas where supercooled rain is present near the surface. Comparing Figure 11a and 11b/c, the area of freezing rain is reduced by approximately 50% in both the F9-HMgc-FFDh and F9-HMgr-FFDl-CB cases relative to F9-BASE (highlighted by black

contours). At the same time, IWC in the western part of the simulation domain increases from approximately 0.1 g m⁻³ in F9-BASE to about 1.0 g m⁻³ in the F9-HMgc-FFDh and F9-HMgr-FFDl-CB cases (as indicated by the white circles in Figures 11d, 11e, and 11f).

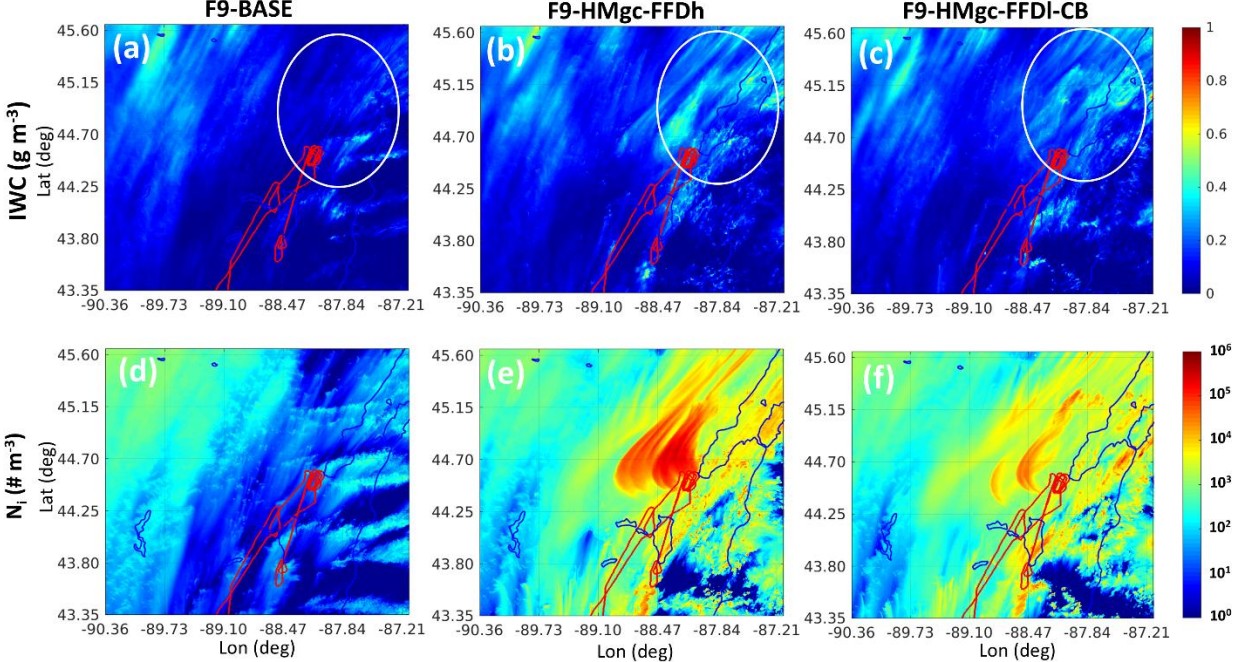

Figure 10. Averaged values between altitude of 0.5 and 3.8 km for Flight 9. (a), (b), (c): IWC; (d), (e), (f): Ni. First column for F9-BASE, middle column for F9-HMgc-FFD, third column for F9-HMgc-FFDl-CB. Red line: flight track.

The reduction in the freezing rain area in the SIP simulations aligns more closely with observations. As shown in Figure 3, the aircraft descended to low altitudes during two simulated missed approaches shortly before and after 20:30 UTC (indicated by the yellow and turquoise arrows respectively). These simulated missed approaches were intentional, pre-planned manoeuvres that replicate a missed landing, allowing the aircraft to collect in-situ observations near the surface. During the first simulated missed approach (Point 1 in Figure 3, marked by the yellow arrow), *in situ* observations indicated the presence of both IWC (Figure 3b) and LWC (Figure 3d). This simulated missed approach corresponds to the yellow-highlighted flight path in Figure 11. All simulations predict the coexistence of liquid and ice, consistent with the observations. In the second simulated missed approach (Point 2 in Figure 3, marked by the turquoise arrow), high IWC values between 0.7 and 1.0 g m⁻³ were observed with no liquid water present, as shown in Figure 3. This second simulated missed approach corresponds to the turquoise-highlighted flight path in Figure 11. In the F9-BASE simulation, both supercooled rain and IWC were predicted, and the IWC value is very low (<0.1 g m⁻³). The F9-HMgc-FFDh and F9-HMgc-FFDl-CB simulations, on the other hand, produced

conditions with no RWC and high IWC, which closely matches the observations. The improved simulation of freezing rain conditions in this study is consistent with the findings of Cholette et al. (2024) in which the freezing rain areas are significantly reduced comparing to the simulation without SIP, despite only using the HM process in their study.

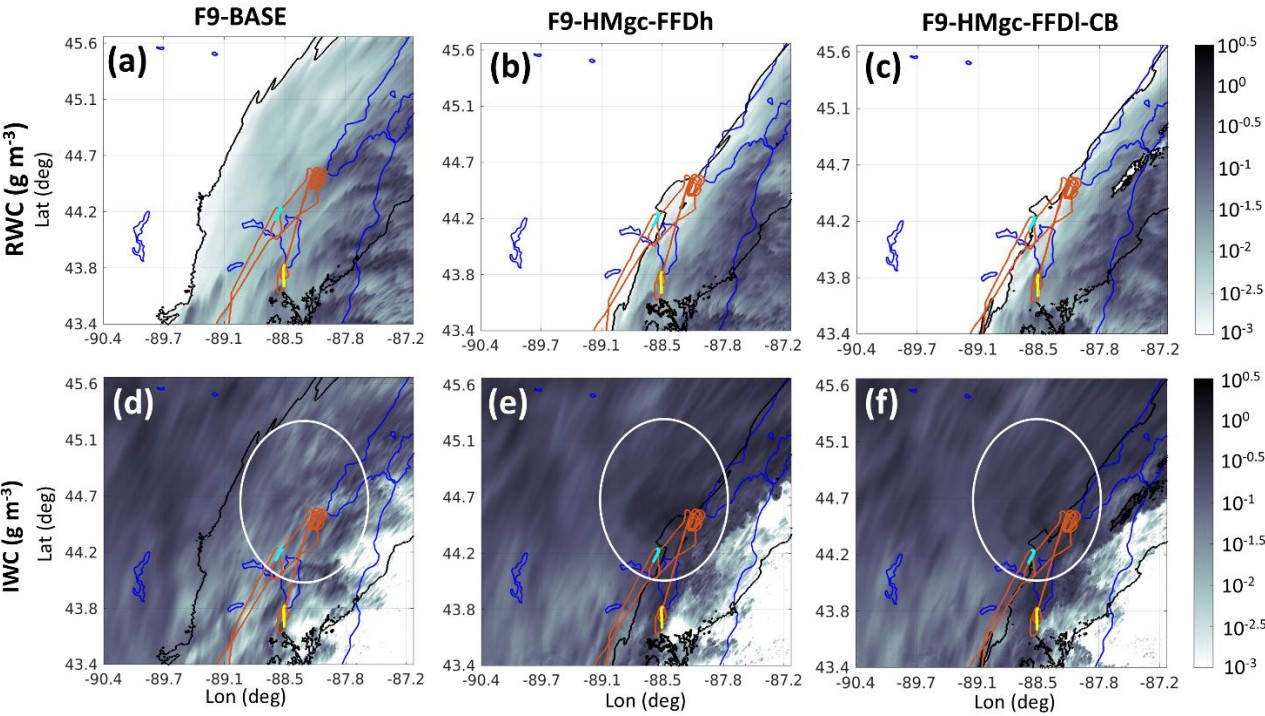

Figure 11. RWC (a, b, c) and IWC (d, e, f) near the surface for F9-BASE (a, d), F9-HMgc-FFDh (b, e) and F9-HMgc-FFDl-CB (c, f) simulations at 20:30 UTC on 2019-02-07. Black contour: region of freezing rain (RWC > 0.01 g m⁻³ and T < 0°C in the lowest model layer at ~20 m above surface). Red lines: Flight 9 aircraft track. Yellow lines: part of flight track below 800 m of altitude observing both supercooled liquid and ice near 20:12 UTC. Turquoise lines: part of flight track below 800 m of altitude observing only ice near 20:47 UTC.

## 5.2. ICICLE Flight 20: stratiform mesoscale case

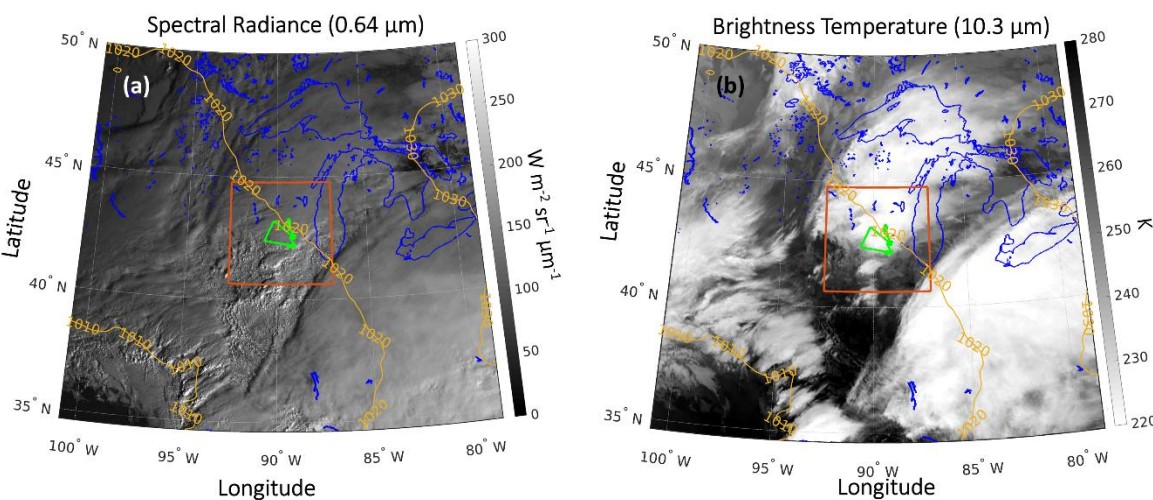

Figure 12. (a) GOES-16 spectral radiance at 0.64 μm (channel 2). (b) GOES-16 brightness temperature at 10.3 μm (channel 13). Both images are from 14:30 UTC on 23 February 2019. Blue lines represent coastlines, green lines represent aircraft flight track, and the red rectangle represents the inner-most simulation domain at 0.25 km resolution for Flight 20.

On 23 February 2019, a significant winter storm impacted the Great Lakes region, including Wisconsin, and northern Illinois, bringing heavy snowfall, high winds, and blizzard conditions. Satellite data from GOES-16 (Figure 12) indicated widespread overcast conditions, with thick clouds dominating the region. The cloud cover consisted of mostly stratiform clouds, typical of large-scale winter systems.

ICICLE Flight 20 was conducted on 23 February, in the area west of Lake Michigan. The flight encountered an extensive cloud system spanning over more than 300 km, within which HIWC conditions were frequently observed. Figure 13 presents the observed X-band radar reflectivity, *in situ* data, and flight altitude. A consistent bright band was detected throughout the flight. A temperature inversion was also observed as the aircraft passed through the bright band. Below the bright band, the aircraft primarily detected liquid water, often supercooled (Figure 13e), with no ice present. Above the bright band, observations indicated mostly ice, often with IWC exceeding 0.5 g m⁻³, and little to no liquid water, particularly between 13:45 and 16:00 UTC. One exception is between 12:25 and 13:45 UTC. As a result, mixed-phase conditions were less frequent in this case compared to the previous Flight 9.

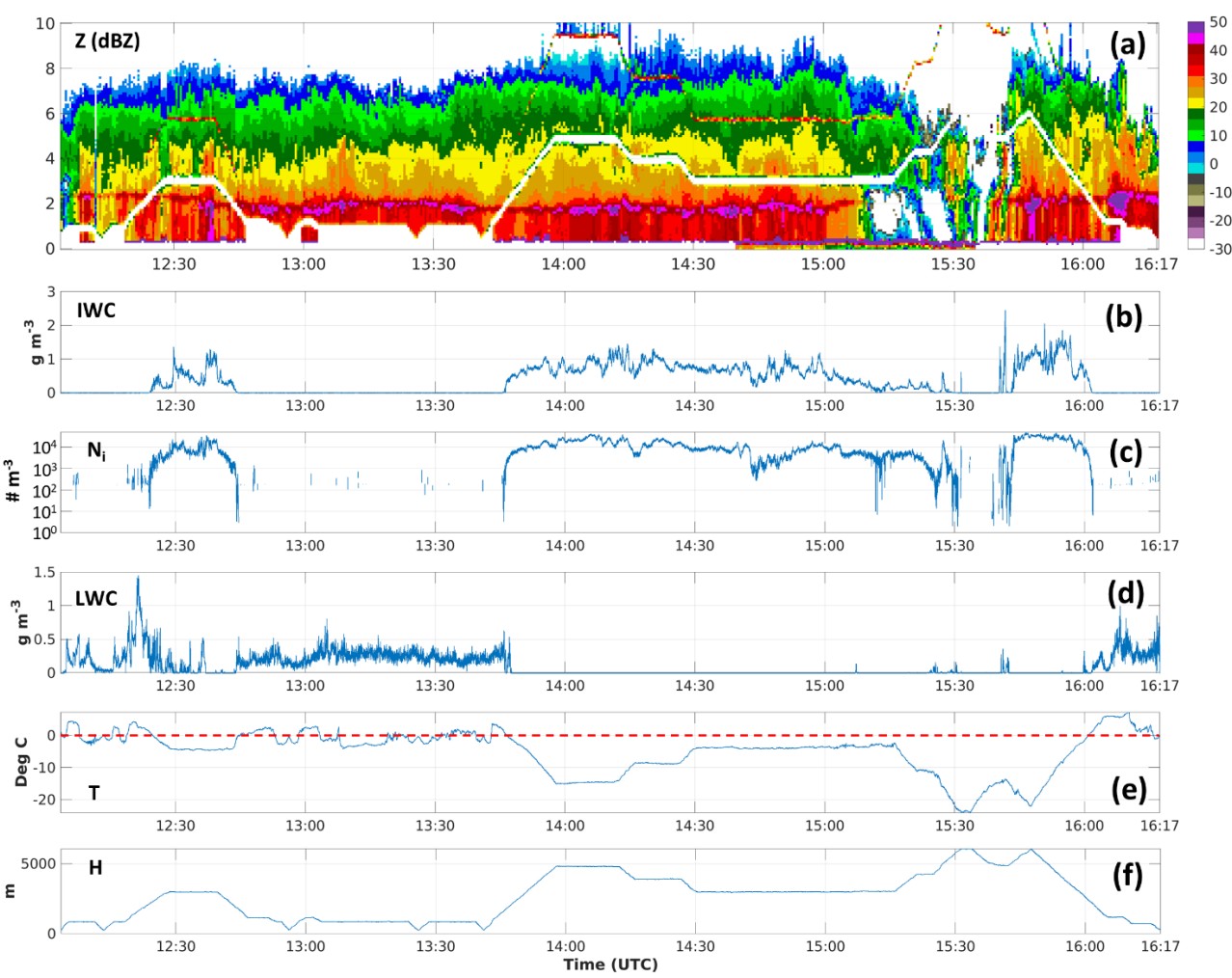

Figure 13. Same as Figure 3 but for ICICLE Flight 20 on 2019-02-23.

Figure 14 shows the simulated brightness temperature at 10.3 μm for the F20-BASE simulations (Figure 14a), along with the observed brightness temperature from GOES-16 on 23 February 2019 at 14:30 UTC (Figure 14b). The F20-BASE simulation effectively reproduces the large-scale sea-level pressure pattern (shown by yellow lines). The simulated brightness temperature

at 10.3 μm closely matches the GOES-16 observations, exhibiting a warm bias, although it slightly underestimates cloud coverage.

The simulated profiles within 10 km of the aircraft track indicate that the impact of SIP (across all four SIP simulations) on IWC and $N_i$ is minimal (Figures 15a, c). The F20-HMgc-FFDh and F9-HMgc-FFDl-CB simulations show the maximum IWC

increases of approximately 10% at altitudes of 3 km and 3.7 km, respectively, compared to the F20-BASE simulation. The
increase in $N_i$ is about 2-3 times, which is significantly smaller than the Flight 9 case, where increases reached 2-3 orders of
magnitude. Additionally, both F20-HMgc-FFDh and F9-HMgc-FFDl-CB generate lower LWC and RWC above 2.5 km
(Figures 16a, b). Due to the higher $N_i$, the values of $Z_{median}$ in both F20-HMgc-FFDh and F9-HMgc-FFDl-CB simulations are
about 3 dBZ lower than in the F20-BASE simulation between 2 and 4 km altitude. No SIP activity is detected in any of the
simulations at 14:30 UTC (Figures 16d to 15g) within 10 km of distance to the flight track, except for F9-HMgc-FFDl-CB,
which exhibits very low rates of both FFD and CB type I. Note that there are regions with SIP activities at this time step, but
they are away from the selected regions for statistics. The simulated case on 23 February 2019 appears more stratiform, with
maximum vertical updraft velocities typically below 4 m s$^{-1}$. This condition likely contributes to the less frequent occurrence
of mixed-phase conditions near the bright band, whereas precipitation size drops driven by updrafts through the melting layer
could play a crucial role in enhancing SIP activity, as suggested by Korolev et al. (2024).

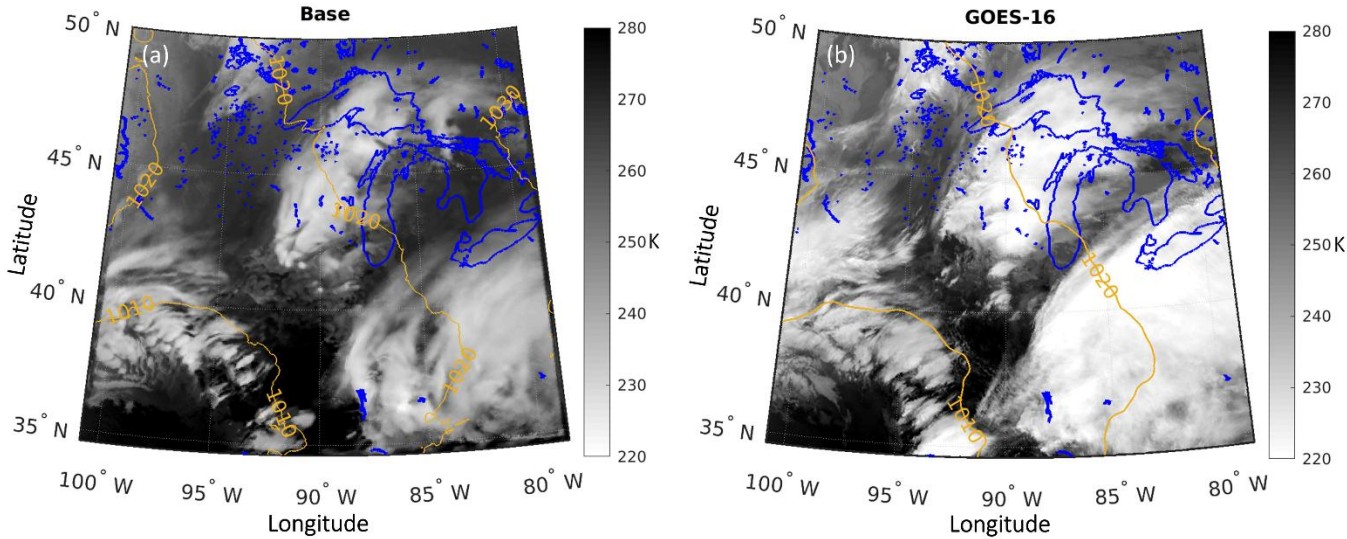

Figure 14. Similar to Figure 4 but for ICICLE Flight 20 on 23 February 2019 at 14:30 UTC. The yellow lines in panel b
indicate the sea-level pressure from ECCC's operation regional prediction at 14:30 UTC initiated at 12:00 UTC.

Although SIP activity during ICICLE Flight 20 is minimal in the simulation, a large area of HIWC conditions is observed.
Unlike localised convective cases during the ICICLE Flight 9 or tropical cases (Huang et al., 2022; Qu et al., 2022; Korolev
et al., 2024), the results from ICICLE Flight 20 suggest that HIWC conditions can develop without significant contributions
from SIP processes, particularly in stratiform and long-lasting cloud systems.

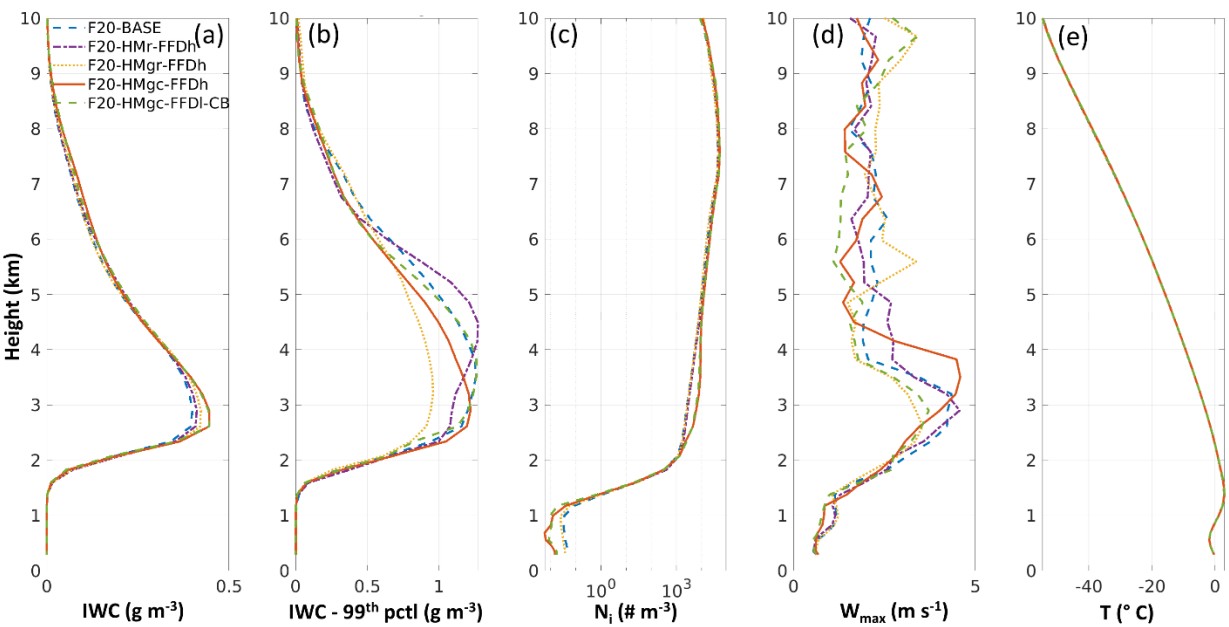

Figure 15. Same as Figure 5 but for ICICLE Flight 20 on 2019-02-23. The profiles are calculated based on simulation data at 510 min (14:30 UTC).

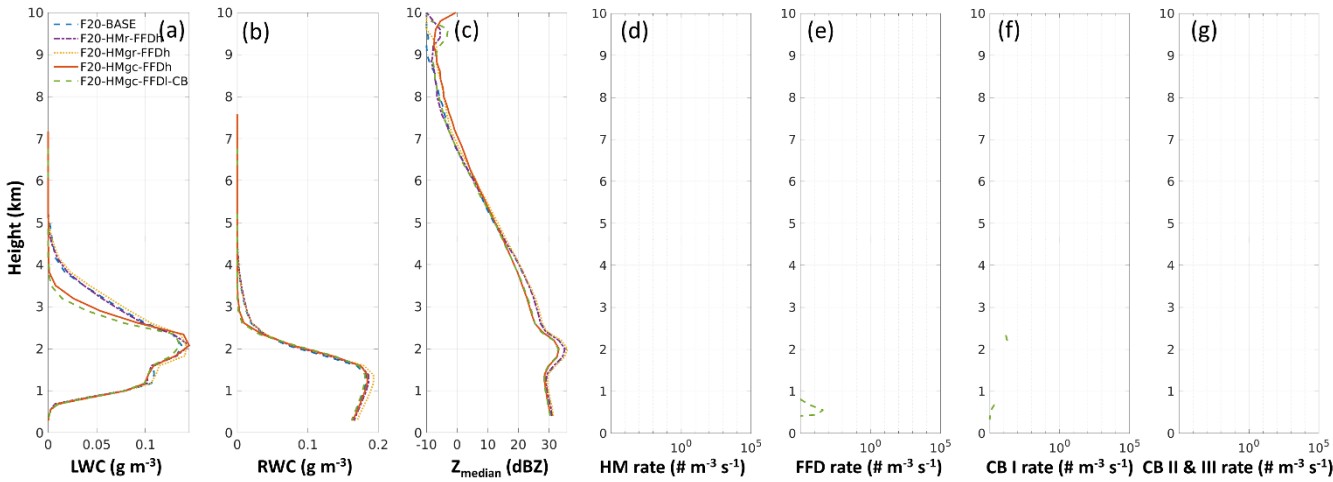

Figure 16. Same as Figure 6 but for ICICLE Flight 20 on 2019-02-23. The profiles are calculated based on simulation data at
510 min (14:30 UTC).

All simulations, including F20-BASE, show reasonably good agreement with the observations for IWC and $N_i$, as illustrated in Figure 17. Unlike the ICICLE Flight 9 case, the F20-BASE simulation does not significantly underestimate the frequency of HIWC conditions. For the $N_i$ distribution, while F20-HMgc-FFDh and F20-HMgc-FFDl-CB provide the closest matches to the observations, the F20-BASE simulation also produces a fairly accurate estimate.

One possible reason for the fairly accurate $N_i$ values in the F20-BASE simulation is that in this stratiform case $N_i$ is likely governed by the heterogeneous nucleation process, with SIP playing a relatively less important role. In the F20-BASE simulation, most $N_i$ values between –15°C and –25°C (T of the maximal flight altitude for Flight 20 at ~6.4 km) are higher than the parameterized $N_i$ due to condensation freezing/deposition ice nucleation (lower half of the small red cycles in Figure 18a) within the same temperature range, but lower than the parameterized $N_i$ for colder temperatures (upper half of the small

red cycles in Figure 18a). Ice particles formed at temperatures below –25°C through condensation freezing/deposition ice nucleation may gradually fall to warmer, lower-altitude regions, thereby contributing to higher $N_i$ compared to the parameterized $N_i$. Another possible source of the enhanced $N_i$ is immersion and contact freezing. Additionally, simulated results from F20-BASE (Figure 18a) show a gradual decrease in $N_i$ with increasing temperature, underscoring the effect of ice particle aggregation.

In the case of Flight 20, the cloud top extends above the homogeneous freezing threshold at -40°C, allowing for the possibility that homogeneous ice nucleation could also influence $N_i$ at lower altitudes as these ice particles descend. However, as shown in Figures 18a, the $N_i$ values above the -40°C level are not substantially higher than most $N_i$ values seen between -20°C and -30°C (ranging from $10^3$ to $10^5$ m$^{-3}$). As ice particles formed through homogeneous freezing descend into warmer temperatures, aggregation processes are expected to further decrease $N_i$, resulting in concentrations lower than those shown between –20 °C

and –30 °C in Figure 18a, where most values range from $10^3$ to $10^5$ m$^{-3}$. This pattern indicates that, at temperatures warmer than -25°C, the ice particle concentrations are more likely dominated by heterogeneous nucleation rather than by the effects of homogeneous nucleation.

Compared to F20-BASE (Figure 18a), F20-HMgc-FFDh (Figure 18b) and F20-HMgc-FFDl-CB (Figure 18c) have more frequent occurrences of higher $N_i$ values for temperatures above –10°C, indicating the influence of SIP. However, these

impacts remain moderate.

In the observed $N_i$ frequency distribution (Figure 18d), two distinct clusters are apparent. One cluster (indicated by the large red circle) aligns closely with the parameterized $N_i$ line, suggesting the influence of condensation freezing/deposition ice nucleation. The second cluster, which exhibits higher $N_i$ values below –20°C, may indicate the influence of immersion and contact freezing, as well as the presence of falling ice particles originating from higher altitudes, regions that were not directly

observed due to the flight's altitude limitations. The higher $N_i$ values may also result from the effects of SIP. However, $N_i$ gradually decreases with increasing temperature. Unlike the F20-HMgc-FFDh (Figure 18b) and F20-HMgc-FFDl-CB (Figure 18c) cases, there is no distinct 'protruding' cluster (green circles in Figure 18b, 18c) that would indicate a sudden $N_i$ increase

at lower altitudes due to SIP. Thus, ice formation in warmer regions (lower altitudes) in Figure 18c is likely dominated by heterogeneous nucleation, although the influence of SIP cannot be ruled out.

Observed $N_i$ values at $T > –20°C$ (Figure 18c, $N_i > 1 \times 10^4$ m$^{-3}$) are frequently higher than those in the simulations (Figures 18a, 18b and 18c, where $N_i$ mostly ranges between $1 \times 10^3$ and $1 \times 10^4$ m$^{-3}$). This discrepancy warrants further investigation into the uncertainties associated with the parameterized heterogeneous nucleation at colder temperatures ($T < –20°C$), the ice–ice collection efficiency and SIP.


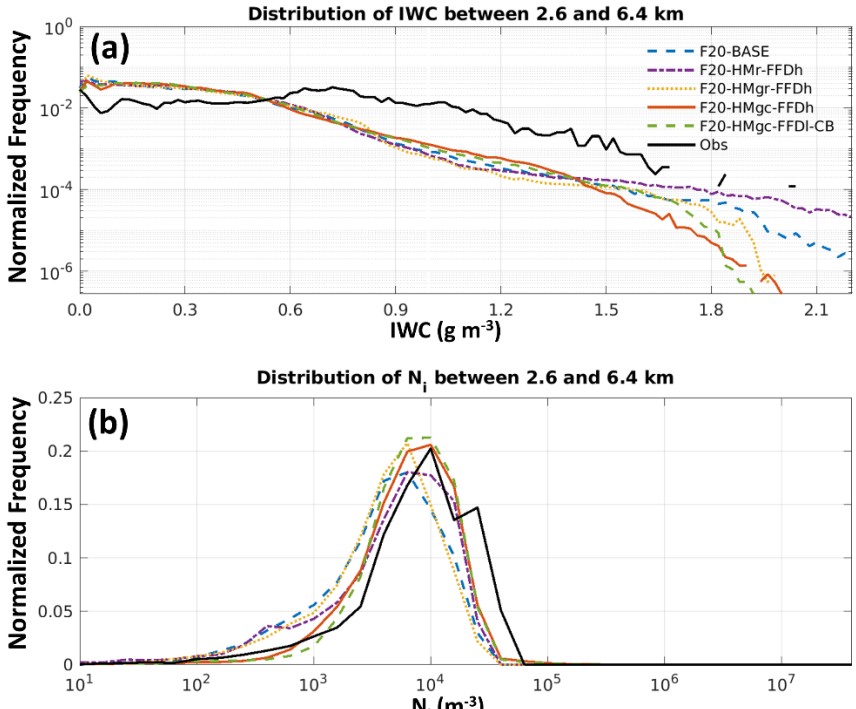

Figure 17. Same as Figure 7 but for simulations at 14:30 UTC on 2019-02-23 and observation data from ICICLE Flight 20 between 12:30 and 16:00 UTC. The altitude range is between 2.6 and 6.4 km.

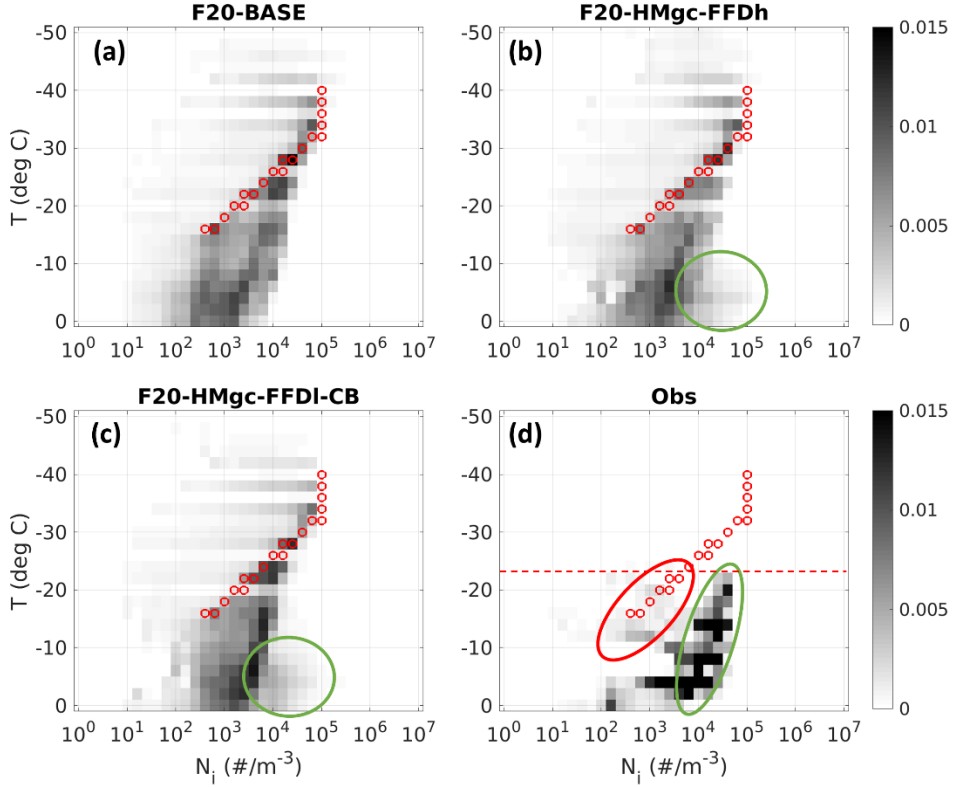

Figure 18. similar to Figure 8 but for Flight 20. Red dashed line: minimal flight T between 12:30 and 16:00 UTC.

For the 2D histogram of Z and the $Z_{median}$ profiles shown in Figure 19, there is good agreement in $Z_{median}$ between the observation and the F20-BASE, F20-HMgc-FFDh and F20-HMgc-FFDl-CB simulations between 2 and 4 km altitude. However, all three simulations underestimate $Z_{median}$ above 4 km.

Several factors may contribute to the underestimation of $Z_{median}$ at higher altitudes, including uncertainties in the initial conditions of the simulation and the exclusion of certain SIP processes. One additional key source of uncertainty is related to the parameterized collection efficiency amongst ice particles. Among the available parameterizations, the range varies by nearly two orders of magnitude between the highest (Lin et al., 1983) and lowest (Ferrier et al., 1995) estimates (Khain and Pinsky, 2018). The current study employs the parameterization from Cotton et al. (1986), which represents a mid-range approach.

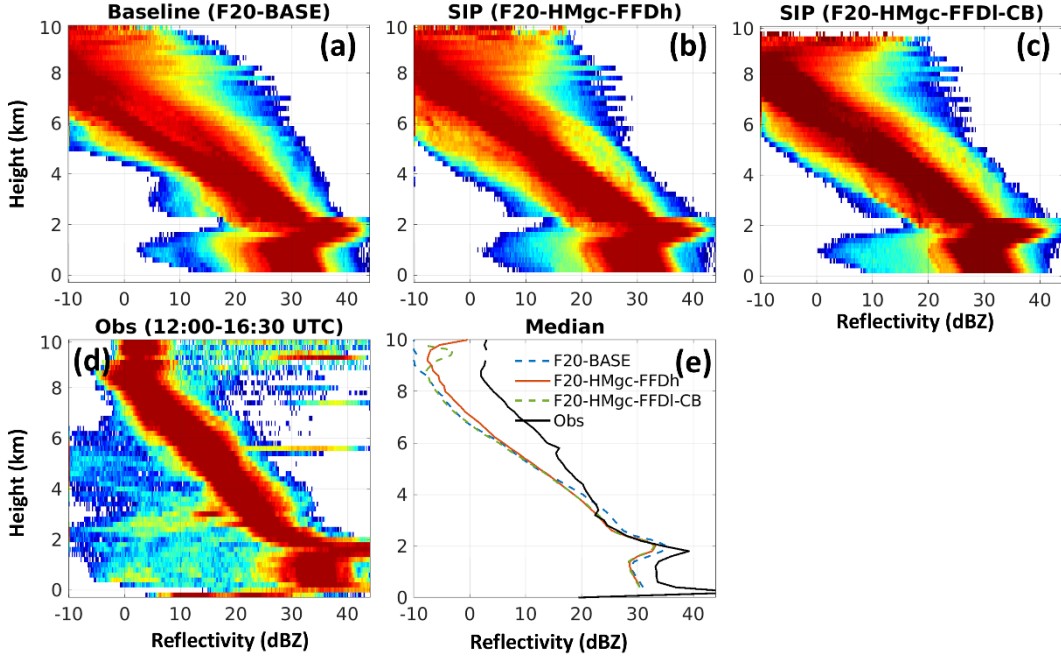

Figure 19. Similar to Figure 9 but for ICICLE Flight 20. The simulation data at 14:30 UTC on 2019-02-23 are used. The observation data from Flight 20 between 12:00 and 16:30 UTC are used.

Figure 20 shows the averaged IWC and $N_i$ between 2.6 and 6.4 km for the F20-BASE, F20-HMgc-FFDh and F20-HMgc-

FFDl-CB simulations. Although the overall differences among the three simulations are relatively small, minor variations in IWC and $N_i$ are noticeable, particularly in localized areas in the western part of the domain. These localized enhancements in IWC and $N_i$ in the two SIP simulations correspond to the locations of updrafts, albeit weaker in this case. The impact on domain-averaged IWC and $N_i$ remains small. Although Figure 16d to 16g show no SIP activity (except very low rates from F20-HMgc-FFDl-CB) at this specific time step within 10 km of distance to the flight track, the local enhancements of IWC

and $N_i$ in close regions to the flight track are the results of SIP processes from earlier time steps.

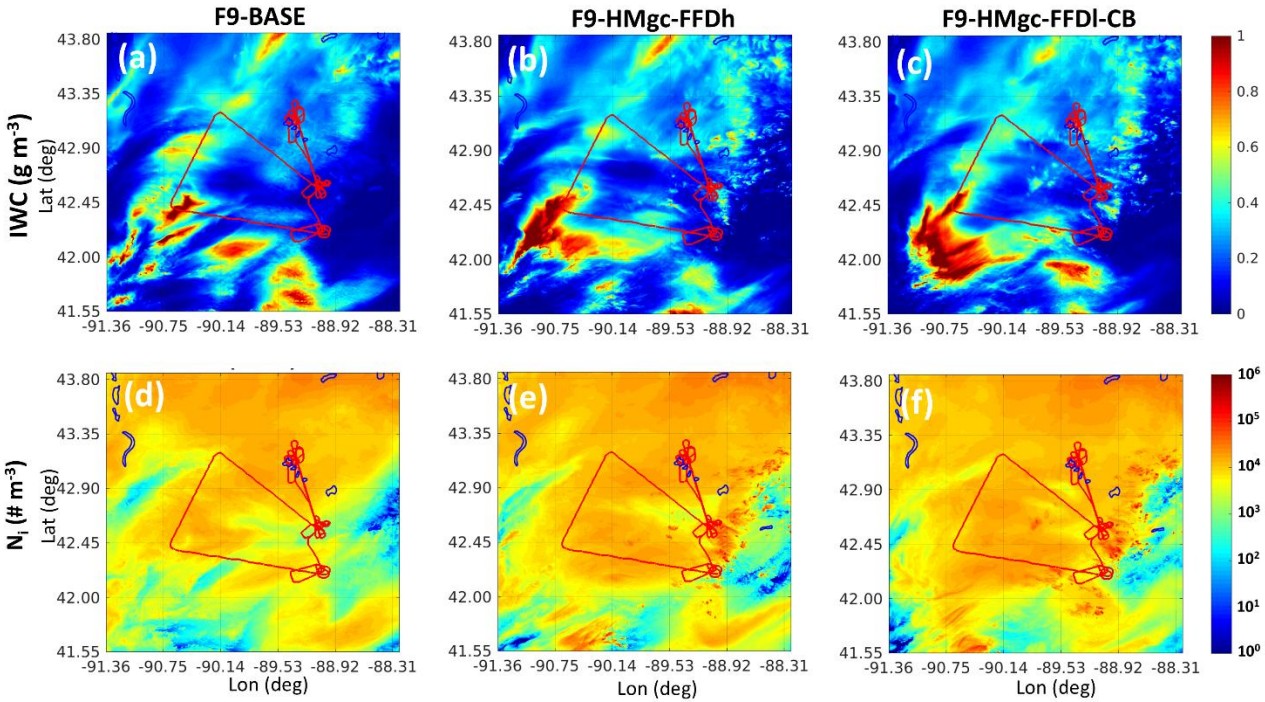

Figure 20. Similar to Figure 10 but for ICICLE Flight 20. The averaging altitudes are between 2.6 and 6.4 km. The time of the simulation is at 14:30 UTC

## 6. Ice particle size distribution for tropical and mid-latitude winter cases

The two mid-latitude winter cases in this study are distinct from each other and also differ from the tropical convection cases previously studied in Huang et al. (2022), Qu et al. (2022) and Korolev et al. (2024). In tropical convection, formation of secondary ice is usually associated with convective updrafts. In mid-latitude winter convective cases, such as ICICLE Flight 9, SIP processes can still significantly influence cloud ice properties, leading to enhanced IWC and $N_i$. In contrast, the mid-latitude winter stratiform case, like ICICLE Flight 20, exhibits very different behavior. Due to the absence of strong convective updrafts and, consequently, fewer instances of mixed-phase conditions, the impact of SIP is minimal. Nevertheless, large areas of HIWC are still observed in this case. This suggests that SIP is neither a necessary nor a sufficient condition for HIWC formation in general. However, SIP is a forcing element which contributes to the enhancement of HIWC and an increase in its longevity (Korolev et al., 2024).

To better understand the differences among these three situations, Figure 21 shows the mass distribution of ice particles by diameter for all three cases for the temperature range between -15°C and -5°C. The solid line represents data from all Convair-580 flights during the HAIC-HIWC campaign in French Guiana (Strapp et al., 2021). Significant SIP activity was observed in

these tropical cases, where most of the ice mass is concentrated around 300 μm, with a reduced fraction of large particles above 1500 μm. The ICICLE Flight 9 case (dashed line) shows a similar distribution, with peak mass centered around 400

μm, but with a higher fraction of ice particles larger than 1500 μm compared to the tropical cases. In contrast, the ICICLE Flight 20 case (dotted line) exhibits an even larger peak mass near 600 μm and a greater fraction of mass located above 800 μm. These results suggest that the ice particle mass and size distribution may be influenced by several factors, including SIP activity, the strength of convective updrafts, and the longevity of the cloud system.

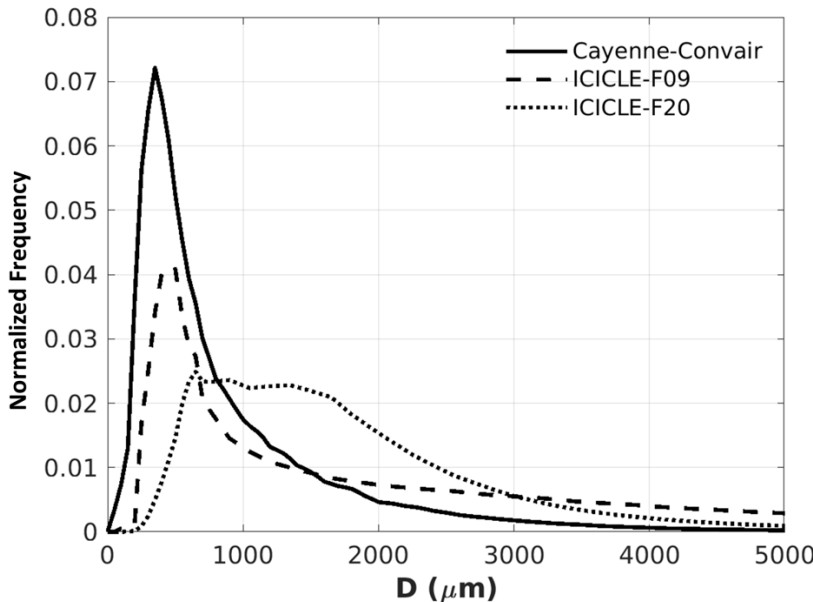

Figure 21. Observed mass distributions as a function of ice particle size (D) from Convair-580 aircraft for HAIC-HIWC campaign near French Guiana (solid line), ICICLE Flight 9 (dashed line) and ICICLE Flight 20 (dotted line) for the temperature between -15°C and -5°C.

In the Cayenne case (tropical setting), the storm system exhibits much stronger convection, with more intense turbulence and updrafts, which in turn enhances SIP processes. For instance, vigorous updrafts and abundant supercooled liquid water near

the melting layer increase the mixing of liquid droplets and ice particles, thereby boosting both HM and FFD rates. Turbulence further amplifies the efficiency of CB. Overall, the $N_i$ in the tropical case (Cayenne) is higher than in the mid-latitude winter cases (ICICLE F9 and F20), resulting in a size distribution that is more heavily weighted toward smaller particles for Cayenne.

Strong updrafts in tropical mesoscale systems limit the growth time of ice particles, as many are rapidly transported to higher altitudes. While these vigorous updrafts can promote the formation of rimed particles such as graupel and hail, such particles

are generally confined to localized regions within the storm, and the majority of ice in tropical systems consists of smaller ice particles. In contrast, ice particles in mid-latitude winter nimbostratus clouds tend to have longer residence times, allowing for

more favorable conditions for depositional growth and aggregation. As a result, aggregation processes are more dominant in mid-latitude winter storms, leading to a greater abundance of larger snow aggregates.

Note that although the fraction of smaller ice particles in the Cayenne case is higher than in ICICLE F20, the total IWC in the Cayenne case is actually greater, with some extreme values exceeding 3 g m$^{-3}$, levels not observed in ICICLE F20.

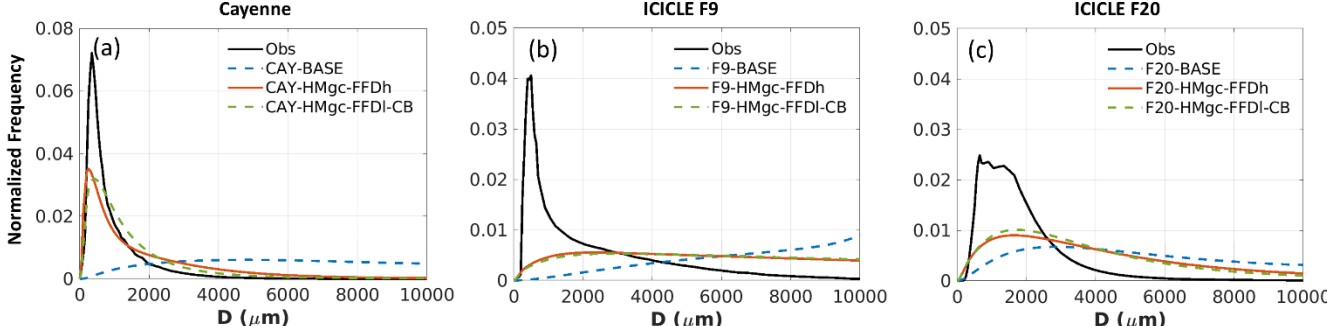

Figure 22. Ice particle mass distribution for HAIC-HIWC 2015 campaign near French Guiana (a), ICICLE Flight 9 (b), Flight 20 (c) for the temperature between -15°C and -5°C. Blue dashed lines: baseline simulation (BASE). Red lines: SIP (HMgc-FFDh) simulation. Green dashed lines: SIP (HMgc-FFDl-CB). Black lines: observation.

Although the impact of SIP processes (HM and FFD) on cloud ice properties varies significantly depending on cloud conditions, such as in ICICLE Flight 9 versus ICICLE Flight 20, incorporating these SIP processes into the P3 microphysics scheme consistently improves the representation of ice particle size distribution. This is evident for the tropical case (Figure 22a), ICICLE Flight 9 (Figure 22b), and ICICLE Flight 20 (Figure 22c). Without SIP, the BASE simulations tend to produce a larger fraction of ice with larger particle sizes.

However, even with SIP, the PSD is not yet perfectly captured. In addition to the SIP processes parameterized in this study (HM, FFD and CB), at least four other known mechanisms are not included: ice fragmentation due to thermal shock (e.g., Dye and Hobbs, 1968), fragmentation of sublimating ice (Oraltay and Hallett, 1989), activation of ice-nucleating particles (INPs) in transient supersaturation zones around freezing drops (Prabhakaran et al., 2020), and break-up of freezing water drops upon impact with ice particles (James et al., 2021). The omission of these additional processes may further contribute to the underestimation of $N_i$, leading to an underrepresentation of smaller ice particles and an overestimation of larger ones. Even among the three parameterized SIP processes considered in this study: HM, FFD, and CB, substantial uncertainties remain. Improved quantification of these processes through laboratory work will be invaluable for reducing these uncertainties.

Heterogeneous ice nucleation may also contribute to underestimation. In the current model with P3 scheme, heterogeneous nucleation is parameterized following Cooper (1986), which assumes a background aerosol environment and provides a generalized temperature-INP relationship. This approach does not explicitly account for variations in INP type, concentration, size distribution, or composition, introducing additional sources of uncertainty into the modelled heterogeneous ice nucleation.

The combined effects of these missing processes and parameterization uncertainties likely contribute to discrepancies between the simulated and observed ice crystal size distributions. Further research is needed to improve our understanding of these issues and to enhance the accuracy of model predictions.

### 7. Conclusions and perspectives


This study examined the impact of three SIP mechanisms, HM, FFD and CB, on two mid-latitude winter cloud cases. For the localized convective case (ICICLE Flight 9), where more vigorous updrafts are present, SIP significantly enhances IWC and $N_i$ while reducing LWC, RWC, and $Z_{median}$. Activating parameterized SIP processes in the simulations results in cloud properties that compare more closely with observational data. In contrast, for the stratiform condition observed in ICICLE

Flight 20, the impacts of SIP are much less pronounced, although the inclusion of SIP still provides slight improvements in the simulated cloud properties.

For localized convective cases like ICICLE Flight 9, which had a temperature inversion and freezing rain, the outcomes of SIP are highly sensitive to how different SIP mechanisms are parameterized. In particular, the ice splinters generated by HM or CB process could trigger the FFD process. Under freezing rain conditions, where there is an abundance of supercooled

raindrops, reaching a critical amount of ice particles can trigger the FFD process in a chain-reaction-like manner, leading to unrealistically high $N_i$ values. Therefore, accurately quantifying each SIP mechanism is essential for better simulating the cloud properties. The recent laboratory study by Seidel et al. (2024), which found no strong evidence for the efficiency of the rime-splintering mechanism, which is in contrast to past studies including Hallet and Mossop, 1974, thus underscoring the need for a better understanding and more precise quantification of SIP processes.

Heterogeneous ice nucleation plays a key role in regulating ice cloud properties under mid-latitude winter conditions. It not only provides the initial ice particles needed to initiate the HM and CB processes, but in stratiform cloud cases like ICICLE Flight 20, where the impact of SIP processes is moderate, it might have a more direct influence on controlling $N_i$. Under these conditions, accurate modeling of the heterogeneous nucleation process is crucial, as current simulation results suggest an underestimation of $N_i$ for T > –20°C during ICICLE Flight 20.

This study also demonstrated that incorporating SIP in NWP models can significantly improve the prediction of freezing rain conditions, which is especially important in mid-latitude regions with severe winter weather. Although, in this study, the simulation results at 250 m grid spacing were primarily discussed, similar conclusions were also found for simulations at other spatial resolutions, specifically at 1 km and 2.5 km grid spacing, which are more commonly used in operational high-resolution NWP systems. Without SIP, the numerical model might overestimate the extent of freezing rain, leading to higher false alarms

and unnecessary public safety concerns. Consistent findings have been reported in other studies, such as Cholette et al. (2024), which showed that including the HM process alone in a 2.5 km NWP system can notably improve freezing rain forecasts.

Thus, integrating SIP processes into operational systems holds promise for enhancing the accuracy of simulated cloud and precipitation properties.

Aviation authorities recognize HIWC environment as one of the hazards for operation of commercial aviation leading to engine power loss, stalls, or damage (e.g., Lawson et al. 1998; Mason and Grzych, 2011, Brawin and Strapp 2019). Knowledge of ice particle size distributions in HIWC environment are important for aircraft engine design to mitigate their icing. The present study suggests PSDs in HIWC conditions may be significantly different compared to those observed in tropics (Strapp et al., 2021). This conclusion is also consistent with the *in situ* observations of PSDs in HIWC in Rugg et al. (2022). Altogether, this indicates the presence of regional and seasonal variations in HIWC microphysical parameters.

Finally, while simulations incorporating the activated SIP process showed improvements relative to observations, this does not imply that the current HM, FFD and CB parameterizations accurately represent the underlying physical processes. Although these formulations draw from laboratory experiments or combined modeling and observational studies, they remain largely *ad hoc*. This research emphasizes the importance of these processes in mid-latitude winter cloud systems and highlights the need to incorporate them into numerical models. However, developing parameterizations that truly capture the detailed physics, beyond just bulk effects, remains to be topics of research.

### Code and Data Availability

*In situ* data are available from the Earth Observation Laboratory (EOL) archive 1265 https://doi.org/10.26023/PSC2-TTQS-390A. The GEM code (version 5.1.0-rc3) is available from https://github.com/ECCC-ASTD-MRD/gem/tree/5.1.0-rc3 (Environment and Climate Change Canada, 2020). The code for the P3 microphysics scheme used is available at https://github.com/P3-microphysics/P3-microphysics (Environment and Climate Change Canada and National Center for Atmospheric Research, 2023). Configuration files to reproduce the GEM simulations are available upon request.

### Authors Contribution

ZQ, AK and JM conceptualized the research goals and aims. ZQ, JM and AK designed the experiments. AK, IH, MW, CN collected and processed ICICLE data. ZQ performed the simulations and analysis with the help from AK, JM, IH, MW and CN. ZQ prepared the manuscript with contributions from all co-authors.

### Competing interests

The authors declare that they have no conflict of interest.

## Acknowledgement

The authors thank Manon Faucher (ECCC) for her help with setting up the GEM model.

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
