# Peer review of "The impacts of secondary ice production on the microphysics and dynamics of mid-latitude cold season convection"

_EGUsphere, 2025_

## Referee Comment (RC1)

This study (Qu et al.) investigates the impact of secondary ice production (SIP) on the cloud microphysics and dynamics under mid-latitude winter conditions based on numerical weather prediction (NWP) simulations. Two flights from the 2019 In-Cloud ICing and Large-drop Experiment (ICICLE) field campaign were selected for model evaluations. SIP processes, the Hallett-Mossop (HM) and fragmentation of freezing drops (FFD), are parameterized and examined in the model simulations. The study finds that SIP is important for high ice water content (HIWC) production in strong convective conditions, whereas it has a less impact in stratiform conditions. The interaction between the two SIP mechanisms is also important for the modeled cloud microphysics.

**General comments**

The results of contrasting the importance of SIP for convective and stratiform conditions through comparison with *in situ* ICICLE observations are interesting. However, the manuscript can be significantly improved by considering the following points:
1. Description of cloud microphysics in model parameterizations be improved (see detailed comment below).
2. Primary ice nucleation and role of it for model simulations.
3. Role of other SIP mechanisms (e.g., ice-ice collision breakup) for model simulations.

**Specific comments**

Line 26: "influencing the energy balance and hydrological cycle on both regional and global scales". Please provide some citations, e.g., Zhao and Liu GRL 2021, among others.

Line 92: "(PSD)"?

Line 99: "sizes of ice and liquid particles, ice and liquid water content (LWC) were calculated from PSDs measured by particle probes". How do you separate liquid from ice particles to calculate LWC and IWC?

Line 102: "asses"? should be "assess".

Figure 1: "The red rectangle". Shall it be "orange rectangle"?

Section 3.2. Description of cloud microphysics parameterizations should be improved.

How is ice nucleation parameterization represented in the model? how about aerosols and CCN treated in the model?
Line 130: "four ice categories". What are these four ice categories?
Line 131: "PSD". No need to redefine it as it is defined above.
Line 133: "liquid ice mass"?
Line 156: "Liquid "cloud droplets" are defined as having a diameter smaller than 50 μm, while "rain drops" have a diameter exceeding this threshold". How does P3 treat liquid-phase hydrometeors in P3?

Equation (3): What are the units of these variables?

Line 180. Considering the large uncertainty of the FFD parameterization, can you test another parameterization Phillips et al. 2018?

Line 186: model experiments "one with only HMr (based on ice collected rain drops, excluding collected rain used for FFD), one with only FFD". These two experiments have never been discussed later and listed in Table 1.

Line 211: "ice number concentration (Ni)". Can you give the size range of ice?

Figure 3: The unit of Ni seems not correct. Same for Ni in Figure 12.

Line 259-261. "These results indicate that under freezing conditions with an abundant supply of supercooled rain, the initiation of the FFD process is highly dependent on the rate of the HM process, or potentially any SIP processes which could provide sufficient amount of small ice particles." It is interesting to see the interactions between SIPs. How about SIP and primary ice nucleation interactions?

Line 266: "based at simulation 510 min simulation time"? The sentence is not correct (two simulations).

Line 390: "in this stratiform 390 case Ni is primarily governed by the homogeneous freezing process, with SIP playing a relatively less important role. Ice particles with high number concentrations, formed through homogeneous freezing above 7 km, fall and eventually increase Ni values at lower altitudes." Can you provide evidence? how about heterogeneous ice nucleation?

Line 450: "However, even with SIP, the PSD is not yet perfectly captured. Further studies are needed to enhance the accuracy of ice particle size distribution representation." Can you give some explanations?

Conclusions and perspectives section: please add some discussions on primary ice nucleation and other SIP processes and role of them for model simulations.

---

## Referee Comment (RC2)

**Egusphere-2025-649 Review**

The manuscript evaluates the impacts of certain SIP parameterizations on cloud microphysical representation in NWP models during mid-latitude storms over the continental US. The authors compare in situ observations from a couple of flights during the ICICLE field campaign with the Global Environmental Multiscale (GEM) model. The results show that HM, FFD SIP parameterizations significantly improve the simulated cloud ice number concentrations and ice water content compared to the base model without SIP. The authors also observed that High Ice Water Content (HIWC) in clouds can occur in midlatitude systems without strong convective conditions and obvious impacts from SIP.

**General Comments**

This is a well-written manuscript with clear objectives, descriptions of the observations and comparison of observations with model results. I also appreciate the effort to explain and address common pitfalls of FFD parameterizations on lines 170-182. The conclusions are generally supported by the observations and analysis presented. I have some minor revisions and comments, which I hope will clarify some missing details and add some much-needed discussion to the paper.

**Minor comments**

1. While the authors focus on the effects of the HM and FFD processes on ice number the IWC in clouds, can other SIP mechanisms such as ice-ice collisions and fragmentation also contribute/complement the HM and FFD processes? For example, Zhao and Liu (2021) showed that the addition of SIP process parameterization such as ice-ice collision fragmentation and droplet shattering during freezing can increase global ice water path by 20% in the CAM6 model. Is it possible that including other SIP process parameterizations may change the result that the F9-HMgc-FFD performs best compared to the BASE, and HMgr-FFD for Flight 9?

2. Line 390: The Flight 20 case is a little confusing. The authors mention that the system primarily consisted of stratiform clouds, but Fig. 12 suggests a deeper system with cloud top heights near 8-9 km above ground. Also, the method of ice formation is claimed to be primarily homogenous freezing and precipitation of ice particles from above. It isn't clear from Fig 12 radar data if such ice precipitation is

observed. Could the authors provide more details regarding the claim of Ni primarily from precipitating ice vs heterogeneous freezing within the storm system?

3. Line 425: Can the authors expand on the reasoning why storm system longevity might contribute to HIWC? Does a growing ice particle feel the storm system lifetime? At first thought, it should fall out of the cloud whenever it gets too heavy, and stronger updrafts should help it stay afloat longer and get bigger. This suggests that stronger updrafts may result in larger ice particles, yet Fig 19 suggests that larger ice particles were observed during weaker updraft conditions. Some explanation/clarification will better help the readers since these are very interesting results.

4. Line 435: Could HIWC be influenced by higher and lower aerosol number concentrations during the French Guiana, ICICLE F09 and F20 flights? Could aerosol cleansing during a longer storm increase supersaturation fluctuations to form bigger ice particles during F20?

**Minor corrections**

Line 113: How thick is the observed melting layer for F09 and F20? What is the model vertical resolution and is it able to resolve the melting layer for each grid size?

Line 149: Is this a typo? The HM peak is at -5 C, and also goes to 0 at -5 C, Should it be -8 C?

Line 164: What is numerical value with units of the max Nf rate at T = -12.5 C?

Line 173: The "which collect rain droplets" makes the sentence a little difficult to read.

Fig 2: Can the F09 track also be overlayed on this figure? It will help the reader locate flight observations, the storm system and model grid centers better.

Flight 224: "The simulated brightness temperature closely matches the GOES-16 observations." With the naked eye, the GOES-16 temperature looks lower for the cloud system. Could you provide mean temp values for both for the cloud system?

Line 304: It isn't clear exactly where the enhancement is occurring in the figure. Could you add a box/arrow to point at the regions?

Line 315: Also unclear where the increase in IWC is to be seen. The scale is massive and the difference in shades of grey is tough to spot.

---

## Author Comment (AC1)

**The impacts of secondary ice production on the microphysics and dynamics of mid-latitude cold season convection**

**Reviewer #1**

**General comments:**

The results of contrasting the importance of SIP for convective and stratiform conditions through comparison with *in situ* ICICLE observations are interesting. However, the manuscript can be significantly improved by considering the following points:

1. Description of cloud microphysics in model parameterizations be improved (see detailed comment below).
2. Primary ice nucleation and role of it for model simulations.
3. Role of other SIP mechanisms (e.g., ice-ice collisional breakup) for model simulations.

   *Thank you very much for your valuable comments which helped to improve the manuscript! Please see the answers below for the specific points (responses in blue).*

**Specific comments:**

Line 26: "influencing the energy balance and hydrological cycle on both regional and global scales". Please provide some citations, e.g., Zhao and Liu GRL 2021, among others.

   *Thank you for the suggestion. Multiple citations have been added into the revised version, including* Field et al. 2017, Kanji et al. 2017, Korolev and Leisner, 2020, Zhao and Liu, 202.

   *We also added this citation into the reference list:*

   *Zhao, X. and Liu, X.: Global importance of secondary ice production. Geophys. Res. Lett., 48, e2021GL092581. https://doi. org/10.1029/2021GL092581, 2021.*

Line 92: "(PSD)"?

   *The original text in the manuscript is:*

   *"The DMT Precipitation Imaging Probe (PIP: Baumgardner et al., 2001) provided measurements of particles in the nominal size range from 100 $\mu$m to 6.4 mm (PSD)."*

   *We changed it to:*

   *"The DMT Precipitation Imaging Probe (PIP: Baumgardner et al., 2001) provided measurements of particle sizes and their concentrations in the nominal size range from 100 $\mu$m to 6.4 mm."*

Line 99: "sizes of ice and liquid particles, ice and liquid water content (LWC) were calculated from PSDs measured by particle probes". How do you separate liquid from ice particles to calculate LWC and IWC?

*For the calculation of LWC and IWC, we used a complex set of rules involving scattering probes (e.g., FCCP, CDP, FCDP), optical array imaging probes (e.g., 2DS, HVPS, PIP), a Rosemount icing detector, and a high-resolution imaging probe (e.g., CPI). The description of the phase discrimination algorithm involves consideration of cloud conditions with different phase compositions and the presence of precipitation-sized liquid drops. This description is quite lengthy and may be too destructive for a reader. For the sake of conciseness, we referenced publications describing the cloud phase discrimination algorithm and the sensitivity of the different cloud microphysical probes to particles with different thermodynamic phases. To address the reviewer's comment, the following statement was added in the text:*

*"The sensitivity of cloud microphysical probes to cloud particle thermodynamic phase and cloud discrimination phase algorithm could be found in e.g., Korolev et al. (2017) and Rugg et al. (2022)."*

*Reference:*

*Korolev, A., G. McFarquhar, P. R. Field, C. Franklin, P. Lawson, Z. Wang, E. Williams, S. J. Abel, D. Axisa, S. Borrmann, J. Crosier, J. Fugal, M. Krämer, U. Lohmann, O. Schlenczek, M. Schnaiter, and M. Wendisch, 2017: Mixed-Phase Clouds: Progress and Challenges. Meteorological Monographs, 58, 5.1–5.50, https://doi.org/10.1175/AMSMONOGRAPHS-D-17-0001.1*

*Rugg, A., A., B. C. Bernstein, J. A. Haggerty, A. Korolev, C. Nguyen, M. Wolde, I. Heckman, and St. DiVito, 2022: High Ice Water Content Conditions Associated with Wintertime Elevated Convection in the Midwest. Journal of Applied Meteorology and Climatology, 61, 559–575, DOI: https://doi.org/10.1175/JAMC-D-21-0189.1*

Line 102: "asses"? should be "assess".

*Thank you! We corrected this typo.*

Figure 1: "The red rectangle". Shall it be "orange rectangle"?

*Because 'red' is used consistently for rectangles or lines of the given color in different figures and no other color could cause confusion, we have decided to continue using 'red' instead of 'orange'.*

Section 3.2. Description of cloud microphysics parameterizations should be improved.

*Thank you for the comments. Please find below the revised section 3.2. Added descriptions are underlined:*

[revised manuscript text omitted]

How is ice nucleation parameterization represented in the model? how about aerosols and CCN treated in the model?

*Thanks for this question! We added additional clarifications in section 3.2. Here are the related descriptions:*

*"Droplet activation is parameterized following the approach of Morrison and Grabowski (2007, 2008), which assumes a constant background aerosol concentration of 300 cm$^{-3}$ with a mean particle size of 0.05 micrometers (μm), composed of ammonium sulfate. Work is currently underway to include prognostic aerosols."*

*"In the "official" P3 scheme, ice is initiated via homogeneous freezing of cloud droplets, immersion/contact freezing of cloud droplets and rain drops (see Morrison and Milbrandt, 2015), heterogeneous nucleation via condensation freezing and deposition (Cooper, 1986), and rime splintering (Hallet and Mossop, 1974). Deposition nucleation is specifically restricted to temperatures at or below 258.15 K and requires ice supersaturation of at least 5%."*

Line 130: "four ice categories". What are these four ice categories?

*Thanks for this comment! We added the following description to for the ice categories:*

*"P3 has a user-specified number of freely evolving generic ice-phase categories, each of whose physical properties evolve continuously in time and space. Because of this, each ice category in P3 can represent any type of ice-phase hydrometeor (with the limitations of a bulk scheme). Also, there is no artificial "conversion" between ice-phase categories, which is an inherent weakness in traditional schemes."*

Line 131: "PSD". No need to redefine it as it is defined above.

*Corrected.*

Line 133: "liquid ice mass"?

*Sorry about the confusion. The revised version does not mention this term anymore.*

Line 156: "Liquid "cloud droplets" are defined as having a diameter smaller than 50 μm, while "rain drops" have a diameter exceeding this threshold". How does P3 treat liquid-phase hydrometeors in P3?

*The description of liquid phase component is added into the updated section of P3:*

*"The liquid-phase component of P3 uses a standard two-category, two-moment approach whereby all liquid-phase hydrometeors are partitioned into "cloud" (non-sedimenting droplets) and "rain", each of whose PSDs are represented by complete gamma functions and each has the mass and number mixing ratios as prognostic (and advected) variables. Droplet activation is parameterized following the approach of Morrison and Grabowski (2007, 2008), which assumes a constant background aerosol concentration of 300 $cm^{-3}$ with a mean particle size of 0.05 micrometers (μm), composed of ammonium sulfate. Work is currently underway to include prognostic aerosols."*

Equation (3): What are the units of these variables?

*We updated the description of the equation as:*

*"where $N_f$ is the statistical average number of ice fragments per drop (#/freezing drop), and D is the freezing liquid drop diameter in μm."*

Line 180. Considering the large uncertainty of the FFD parameterization, can you test another parameterization Phillips et al. 2018?

*We have not tested the parameterization of Phillips et al. (2018). However, a recent study by Lachapelle et al. (2025) found that FFD rates derived from Phillips et al. (2018) tend to be low, while those based on Lawson et al. (2015) may be too high. To address this, Lachapelle et al. proposed modified rates for both parameterizations. In our study, we tested the FFD parameterization of Lawson et al. (2015) incorporating the corrections suggested by Lachapelle et al. The SIP rate for FFD follows the original exponent-4 approach (Eq. 3), but is capped when D exceeds 1878.2 μm. Additionally, a linear temperature dependence is applied, with an actual rate ($N_f$) rate at –12.5°C which will be scaled down ($c*N_f$) to zero at –25°C and –2°C (c is a T dependent linear scaling factor between 0 and 1).*

*In addition to the modified FFD process, we incorporated the ice–ice collisional breakup process (Phillips et al. 2017b) into the simulation. An additional adjustment was made by restricting collisional breakup to cases where the relative velocity between two ice particles exceeds 1 m $s^{-1}$ (Hoarau et al. 2018, Korolev and Leisner, 2020).*

*The new experiment with three SIP processes is named as "HMgc-FFDl-CB". FFDl stands for FFD at lower rates (with the correction proposed by Lachapelle et al. 2025), compared to the original version at higher rates. CD stands for collisional breakup.*

*Here is a revised Fig. 5 (new experiment results in green dashed line) for F9:*

[Figure]

*Figure 5. Profiles from the baseline and SIP simulations on 2019-02-07 for the case of ICICLE Flight 9. (a): mean IWC, (b): 95 percentile of IWC (99th pctl of IWC), (c): mean $N_i$ with the number of ice particle smaller than 40 μm excluded, (d): maximum vertical wind speed ($w_{max}$), (d) T (the three simulations have only slightly different temperature that is not distinguishable in the figure). All profiles are calculated from simulation results at 510 min (20:30 UTC) after the initiation time of 12:00 UTC. All profiles are calculated for the region within 10 km of distance to the aircraft track.*

*The new experiment incorporating all three SIP processes (green dashed lines) yields results similar to the HMgc-FFDh simulation (red lines), which uses the same HM process and a higher FFD rate. We observed that applying the higher FFD rate (original approach, not shown) in combination with both HM and CB processes leads to a significant increase in $N_i$ near the surface, as both HM and CB processes produce large numbers of ice splinters that further enhance FFD. However, such extremely high values of $N_i$ were not observed in the actual data. In the new experiment, the use of the reduced FFD rate prevented this explosive increase in $N_i$ near the surface. This suggests that the FFD rate based on Lawson et al. may be overestimated under this condition, while the reduced rate proposed by Lachapelle et al. (2025) improves the agreement between simulations and observations.*

*We add this paragraph to the manuscript:*

*"Including the CB process in the experiment led to an explosive increase in FFD rates near the surface (not shown). To mitigate this, we applied the FFDl (lower rate) parameterization, as proposed by Lachapelle et al. (2025). Results from the three-process experiment (F9-HMgc-FFDl-CB) were similar to those of F9-HMgc-FFDh in terms of $N_i$, IWC, LWC, and $Z_{median}$. The CB process (types I, II, and III) produced positive SIP rates at multiple altitudes, including levels below 2 km (Figure 6f, 6g), potentially further activating the FFD process. The use of the FFDl parameterization effectively suppressed near-surface FFD rates, resulting in values that were substantially lower and more realistic."*

*Here is a revised Fig. 6 (new experiment results in green dashed line) for Flight 9:*

[Figure]

*Figure 6. Profiles from the baseline and SIP simulations on 2019-02-07. (a): mean LWC, (b): mean RWC, (c): median radar reflectivity (Zmedian), (d): mean Hallett-Mossop rate, (e): mean Fragmentation of Freezing Drop rate, (f): ice-ice collisional breakup rate (type I), (g): ice-ice collisional breakup rate (type II & III) All profiles are calculated for the region within 10 km of distance to the aircraft track and at 20:30 UTC.*

*Similar conclusion for Fig. 6. The panels f and g show the SIP rate from ice-ice collisional breakup for type I (involve graupel and hail only), and for type II & III (involve snow/ice crystal with/without graupel and hail).*

*Here is a revised Fig. 7 (new experiment results in green dashed line) for F9:*

[Figure]

*Figure 7. Distribution of IWC (a) and Ni with the number of ice particle smaller than 40 μm excluded (b) for model simulations at 20:30 UTC on 2019-02-07 and for the observation data for ICICLE Flight 9 between 20:00 and 21:00 UTC. The results from model simulations are calculated for altitudes between 0.5 and 5.5 km and with ice water content higher than 0.001 g m-3 in the region within 20 km of distance to the aircraft track. The results from in situ observation are calculated for the condition with IWC higher than 0.001 g m-3. (b): logarithmic bin width of 1/5 of*

*an order of magnitude is used. Blue dashed lines: Baseline simulation, purple dash-dot line: SIP (F9-HMr-FFDh), yellow dotted lines: SIP (F9-HMgr-FFDh) simulation, red lines: SIP (F9-HMgc-FFDh) simulation, green dashed lines: SIP (F9-HMgc-FFDl-CB), black lines: observation.*

*The new experiment (green dashed lines) produces similar $N_i$ distribution with the cases of extreme high $N_i$ (>2x10$^5$ m$^{-3}$) reduced even compared to HMgc-FFDh (red line).*

*The HMgc-FFDl-CB simulation results have been included in most of the figures including for Flight 20, though they are not shown in this response document. For further details and discussion, please refer to the revised manuscript.*

Line 186: model experiments "one with only HMr (based on ice collected rain drops, excluding collected rain used for FFD), one with only FFD". These two experiments have never been discussed later and listed in Table 1.

*Sorry about the confusion. The two simulations mentioned here were not shown in the initial submitted version. We only intend to show four simulations. We change the description to:*

*"Four additional simulations were carried out: three including both the HM and FFD, and one including HM, FFD and CB. In the three two-process experiments, the same FFDh (higher rate) parameterization is used, coupled with three different HM parameterizations as described in Section 3.3.1. In the final experiment, all three SIP processes were enabled, with the FFDl (lower rate) parameterization applied."*

Line 211: "ice number concentration (Ni)". Can you give the size range of ice?

*The ice particles with size smaller than 40 microns were excluded in the $N_i$ shown in Fig. 3. For better comparisons, we also excluded ice particles smaller than 40 μm in all the model simulated $N_i$ values using their size distributions. The text here was revised as:*

*"ice number concentration ($N_i$, excluding the particle diameter smaller than 40 μm)"*

Figure 3: The unit of Ni seems not correct. Same for Ni in Figure 12.

*The $N_i$ values presented in Fig. 3 appear consistent with the statistical results shown in Fig. 7, with most cases displaying values between $1\times10^{-3}$ and $1\times10^{5}$ $m^{-3}$. Similarly, the results for F20 are also consistent, showing comparable value ranges between those depicted in Fig. 12 and Fig. 16 (which are similar to those observed for F9).*

Line 259-261. "These results indicate that under freezing conditions with an abundant supply of supercooled rain, the initiation of the FFD process is highly dependent on the rate of the HM process, or potentially any SIP processes which could provide sufficient amount of small ice particles." It is interesting to see the interactions between SIPs. How about SIP and primary ice nucleation interactions?

*Thank you for this excellent question! We added an additional discussion on this in the manuscript:*

*"In the simulations for Flight 9, condensation freezing/deposition ice nucleation likely initiated ice particle formation at altitudes above 4.5 km (T<-15°C, as implemented in P3 scheme). As these ice crystals descend, they tend to aggregate, leading to decreased $N_i$ values (typically less than $10^{2}$ $m^{-3}$) (see Figure 7b, blue dashed line from the F9-BASE simulation). In addition, the immersion/contact freezing process could also introduce low number of $N_i$ at higher T (T<-4°C). Ice particles produced via heterogeneous nucleation can subsequently trigger the HM or CB processes. However, since $N_i$ generated by heterogeneous nucleation remains low at lower altitudes, where supercooled liquid drops are abundant, the likelihood of activating significant FFD rates is limited. In contrast, the HM and CB processes can produce $N_i$ values at least an order of magnitude higher than those from heterogeneous nucleation alone, substantially increasing the potential for interactions between ice splinters and supercooled raindrops."*

Line 266: "based at simulation 510 min simulation time"? The sentence is not correct (two simulations).

*There are 4 simulations using different parameterizations of SIP, but the results shown in Fig. 5 are all from the same time step at 510 min after the model initiation at 2019-02-07 12:00 UTC. We noticed a typo in the figure caption which might be confusion. We revised it as:*

*"Figure 5. Profiles from the baseline and SIP simulations on 2019-02-07 for the case of ICICLE Flight 9. (a): mean IWC, (b): 95 percentile of IWC (99th pctl of IWC), (c): mean $N_i$ with the number of ice particle smaller than 40 μm excluded, (d): maximum vertical wind speed (wmax), (d) T (the three simulations have only slightly different temperature that is not distinguishable in the figure). All profiles are calculated from simulation results at 510 min (20:30 UTC) after the initiation time of 12:00 UTC. All profiles are calculated for the region within 10 km of distance to the aircraft track."*

Line 390: "in this stratiform 390 case Ni is primarily governed by the homogeneous freezing process, with SIP playing a relatively less important role. Ice particles with high number concentrations, formed through homogeneous freezing above 7 km, fall and eventually increase Ni values at lower altitudes." Can you provide evidence? how about heterogeneous ice nucleation?

*Thank you for this comment! After further analyzing the data, we changed our claim here, and added discussions below in the manuscript:*

*"One possible reason for the fairly accurate $N_i$ values in the F20-BASE simulation is that in this stratiform case $N_i$ is likely governed by the heterogeneous nucleation process, with SIP playing a relatively less important role. In the F20-BASE simulation, most $N_i$ values between –15°C and –25°C (T of the maximal flight altitude for Flight 20 at ~6.4 km) are higher than the parameterized $N_i$ due to condensation freezing/deposition ice nucleation (lower half of the small red cycles in Figure 18a) within the same temperature range, but lower than the parameterized $N_i$ for colder temperatures (upper half of the small red cycles in Figure 18a). Ice particles formed at temperatures below –25°C through heterogenous nucleation may gradually fall to warmer, lower-altitude regions, thereby contributing to higher $N_i$ compared to the parameterized $N_i$. Additionally, simulated results from F20-BASE (Figure 18a) show a gradual decrease in $N_i$ with increasing temperature, underscoring the effect of ice particle aggregation.*

*In the case of Flight 20, the cloud top extends above the homogeneous freezing threshold at -40°C, allowing for the possibility that homogeneous ice nucleation could also influence $N_i$ at lower altitudes as these ice particles descend. However, as shown in Figures 18a, the $N_i$ values above the -40°C level are not substantially higher than most $N_i$ values seen between -20°C and -30°C (ranging from $10^3$ to $10^5$ m $^{-3}$). As ice particles formed through homogeneous freezing descend into warmer temperatures, aggregation processes are expected to further decrease Ni, resulting in concentrations lower than those shown between –20 °C and –30 °C in Figure 18a, where most values range from $10^3$ to $10^5$ m $^{-3}$. This pattern indicates that, at temperatures warmer than -25°C, the ice particle concentrations are more likely dominated by heterogeneous nucleation rather than by the effects of homogeneous nucleation.*

*Compared to F20-BASE (Figure 18a), F20-HMgc-FFDh (Figure 18b) and F20-HMgc-FFDl-CB (Figure 18c) have more frequent occurrences of higher $N_i$ values for temperatures above –10°C, indicating the influence of SIP. However, these impacts remain moderate.*

*In the observed $N_i$ frequency distribution (Figure 18d), two distinct clusters are apparent. One cluster (indicated by the large red circle) aligns closely with the parameterized $N_i$ line, suggesting the influence of heterogeneous ice nucleation. The second cluster, which exhibits higher $N_i$ values below –20°C, may indicate the presence of falling ice particles originating from higher altitudes, regions that were not directly observed due to the flight's altitude limitations. The higher $N_i$ values may also result from the effects of SIP. However, $N_i$ gradually decreases with increasing temperature. Unlike the F20-HMgc-FFDh (Figure 18b) and F20-HMgc-FFDl-CB (Figure 18c) cases, there is no distinct 'protruding' cluster (green circles in Figure 18b, 18c) that would indicate a sudden $N_i$ increase at lower altitudes due to SIP. Thus, ice formation in warmer regions (lower altitudes) in Figure 18c is likely dominated by heterogeneous nucleation, although the influence of SIP cannot be ruled out.*

*Observed $N_i$ values at T > –20°C (Figure 18c, $N_i$ > 1 × $10^4$ m $^{-3}$) are frequently higher than those in the simulations (Figures 18a, 18b and 18c, where $N_i$ mostly ranges between 1 × $10^3$ and 1 × $10^4$ m$^{-3}$). This discrepancy warrants further investigation into the uncertainties associated*

*with the parameterized heterogeneous nucleation at colder temperatures (T < –20ºC), the ice–ice collection efficiency and SIP."*

[Figure]

*Figure 18: 2D histograms for the frequency of Ni regarding the T for F20-BASE (panel a), F20-HMgc-FFDh (panel b), F20-HMgc-FFDl-CB (panel c), and the observation for Flight 20 (panel d). Small red cycles: parameterized $N_i$ due to condensation freezing/deposition ice nucleation (adapted from Cooper, 1986). Red dashed line: minimal flight T between 12:30 and 16:00 UTC.*

Line 450: "However, even with SIP, the PSD is not yet perfectly captured. Further studies are needed to enhance the accuracy of ice particle size distribution representation." Can you give some explanations?

*Thank you for your question! In addition to the analyses already conducted, we also examined the size distribution for the HMgc-FFDl-CB simulations. We found similar results to those obtained from the HMgc-FFDh-CB simulations, as shown in the updated Fig. 22:*

[Figure]

*Figure 22. Ice particle mass distribution for HAIC-HIWC 2015 campaign near French Guiana (a), ICICLE Flight 9 (b), Flight 20 (c) for the temperature between -15°C and -5°C. Blue dashed lines: baseline simulation (BASE). Red lines: SIP (HMgc-FFDh) simulation. Green dashed lines: SIP (HMgc-FFDl-CB). Black lines: observation.*

*We have several hypotheses for the underestimation of the fraction of smaller ice particles, particular for those smaller than 2000 μm. We added the description below in the manuscript to provide a more detailed explanation:*

*"In the Cayenne case (tropical setting), the storm system exhibits much stronger convection, with more intense turbulence and updrafts, which in turn enhances SIP processes. For instance, vigorous updrafts and abundant supercooled liquid water near the melting layer increase the mixing of liquid droplets and ice particles, thereby boosting both HM and FFD rates. Turbulence further amplifies the efficiency of CB. Overall, the $N_i$ in the tropical case (Cayenne) is higher than in the mid-latitude winter cases (ICICLE F9 and F20), resulting in a size distribution that is more heavily weighted toward smaller particles for Cayenne.*

*Strong updrafts in tropical mesoscale systems limit the growth time of ice particles, as many are rapidly transported to higher altitudes. While these vigorous updrafts can promote the formation of rimed particles such as graupel and hail, such particles are generally confined to localized regions within the storm, and the majority of ice in tropical systems consists of smaller ice particles. In contrast, ice particles in mid-latitude winter nimbostratus clouds tend to have longer residence times, allowing for more favorable conditions for depositional growth and aggregation. As a result, aggregation processes are more dominant in mid-latitude winter storms, leading to a greater abundance of larger snow aggregates.*

*Note that although the fraction of smaller ice particles in the Cayenne case is higher than in ICICLE F20, the total IWC in the Cayenne case is actually greater, with some extreme values exceeding 3 g m$^{-3}$, levels not observed in ICICLE F20."*

Conclusions and perspectives section: please add some discussions on primary ice nucleation and other SIP processes and role of them for model simulations.

*This is a great suggestion! We added the following paragraphs for the discussion:*

*"Heterogeneous ice nucleation plays a key role in regulating ice cloud properties under mid-latitude winter conditions. It not only provides the initial ice particles needed to initiate the HM and CB processes, but in stratiform cloud cases like ICICLE Flight 20, where the impact of SIP processes is moderate, it might have a more direct influence on controlling Ni. Under these conditions, accurate modeling of the heterogeneous nucleation process is crucial, as current simulation results suggest an underestimation of Ni for T > –20°C during ICICLE Flight 20."*

---

## Author Comment (AC2)

**The impacts of secondary ice production on the microphysics and dynamics of mid-latitude cold season convection**

**Reviewer #2**

**General Comments:**

This is a well-written manuscript with clear objectives, descriptions of the observations and comparison of observations with model results. I also appreciate the effort to explain and address common pitfalls of FFD parameterizations on lines 170-182. The conclusions are generally supported by the observations and analysis presented. I have some minor revisions and comments, which I hope will clarify some missing details and add some much-needed discussion to the paper.

> *We deeply appreciate your comments for helping to improve this manuscript! Please see our responses to your questions below (in blue).*

**Minor comments:**

1. While the authors focus on the effects of the HM and FFD processes on ice number the IWC in clouds, can other SIP mechanisms such as ice-ice collisions and fragmentation also contribute/complement the HM and FFD processes? For example, Zhao and Liu (2021) showed that the addition of SIP process parameterization such as ice-ice collision fragmentation and droplet shattering during freezing can increase global ice water path by 20% in the CAM6 model. Is it possible that including other SIP process parameterizations may change the result that the F9-HMgc-FFD performs best compared to the BASE, and HMgr-FFD for Flight 9?

> *Thank you very much for this suggestion. We recently implemented the ice-ice collision break-up (CB) process and tested its impact on the simulation results, following Phillips et al. 2017. We make additional adjustment to the method by limiting the collision breakup only to the situation when the relative velocity between two ice particles is larger than 1 m s$^{-1}$ (Hoarau et al. 2018, Korolev and Leisner, 2020).*

> *For Flight 9, including CB further increases $N_i$ at various altitudes and can potentially lead to an unrealistic surge in ice splinter production near the surface due to the rapid FFD rate. To address this issue, we implemented a correction to the FFD process based on the recommendations of Lachapelle et al. (2025). The SIP rate for FFD follows the original exponent-4 approach (Eq. 3), but is capped when D exceeds 1878.2 μm. Additionally, a linear temperature dependence is applied, with an actual rate ($N_f$) rate at –12.5°C which will be scaled down ($c*N_f$) to zero at –25°C and –2°C (c is a T dependent linear scaling factor between 0 and 1).*

> *The additional simulation with three SIP processes included in this manuscript under the experiment "HMgc-FFDl-CB," where FFDl refers to FFD at lower rates (as opposed to the original higher rate, FFDh). CB denotes ice–ice collisional breakup.*

> *The new experiment incorporating all three SIP processes (green dashed lines) yields results similar to the HMgc-FFDh simulation (red lines), which uses the same HM process and a*

*higher FFD rate. We observed that applying the higher FFD rate (original approach, not shown) in combination with both HM and CB processes leads to a significant increase in $N_i$ near the surface, as both HM and CB processes produce large numbers of ice splinters that further enhance FFD. However, such extremely high values of $N_i$ were not observed in the actual data. In the new experiment, the use of the reduced FFD rate prevented this explosive increase in $N_i$ near the surface. This suggests that the FFD rate based on Lawson et al. may be overestimated, while the reduction proposed by Lachapelle et al. (2025) improves the agreement between simulations and observations.*

*Here is a revised Fig. 5 (new experiment results in green dashed line) for F9:*

[Figure]

*Figure 5. Profiles from the baseline and SIP simulations on 2019-02-07 for the case of ICICLE Flight 9. (a): mean IWC, (b): 95 percentile of IWC (99th pctl of IWC), (c): mean Ni with the number of ice particle smaller than 40 μm excluded, (d): maximum vertical wind speed (wmax), (d) T (the three simulations have only slightly different temperature that is not distinguishable in the figure). All profiles are calculated from simulation results at 510 min (20:30 UTC) after the initiation time of 12:00 UTC. All profiles are calculated for the region within 10 km of distance to the aircraft track.*

*An additional paragraph was added into the manuscript:*

*"Including the CB process in the experiment led to an explosive increase in FFD rates near the surface (not shown). To mitigate this, we applied the FFDl (lower rate) parameterization, as proposed by Lachapelle et al. (2025). Results from the three-process experiment (F9-HMgc-FFDl-CB) were similar to those of F9-HMgc-FFDh in terms of $N_i$, IWC, LWC, and $Z_{median}$. The CB process (types I, II, and III) produced positive SIP rates at multiple altitudes, including levels below 2 km (Figure 6f, 6g), potentially further activating the FFD process. The use of the FFDl parameterization effectively suppressed near-surface FFD rates, resulting in values that were substantially lower than the explosive rates simulated in F9-HMr-FFDh (purple dashed lines)."*

*Similar conclusion for Fig. 6. The panels f and g show the SIP rate from ice-ice collisional breakup for type I (involve graupel and hail only), and for type II & III (involve snow/ice crystal with/without graupel and hail).*

*Here is a revised Fig. 6 (new experiment results in green dashed line) for F9:*

[Figure]

*Figure 6. Profiles from the baseline and SIP simulations on 2019-02-07. (a): mean LWC, (b): mean RWC, (c): median radar reflectivity (Zmedian), (d): mean Hallett-Mossop rate, (e): mean Fragmentation of Freezing Drop rate, (f): ice-ice collisional breakup rate (type I), (g): ice-ice collisional breakup rate (type II & III) All profiles are calculated for the region within 10 km of distance to the aircraft track and at 20:30 UTC.*

*Here is a revised Fig. 7 (new experiment results in green dashed line) for F9:*

[Figure]

*Figure 7. Distribution of IWC (a) and Ni with the number of ice particle smaller than 40 μm excluded (b) for model simulations at 20:30 UTC on 2019-02-07 and for the observation data for ICICLE Flight 9 between 20:00 and 21:00 UTC. The results from model simulations are calculated for altitudes between 0.5 and 5.5 km and with ice water content higher than 0.001 g m-3 in the region within 20 km of distance to the aircraft track. The results from in situ observation are calculated for the condition with IWC higher than 0.001 g m-3. (b): logarithmic bin width of 1/5 of an order of magnitude is used. Blue dashed lines: Baseline simulation, purple dash-dot line: SIP (F9-HMr-FFDh), yellow dotted lines: SIP (F9-HMgr-FFDh) simulation, red lines: SIP (F9-HMgc-FFDh) simulation, green dashed lines: SIP (F9-HMgc-FFDl-CB), black lines: observation.*

*The new experiment (green dashed lines) produces similar $N_i$ distribution with the cases of extreme high $N_i$ (>2x10$^5$ m$^{-3}$) reduced even compared to HMgc-FFDh (red line).*

*Reference:*

*Hoarau, T., Pinty, J.-P., and Barthe, 455 C.: A representation of the collisional ice break-up process in the two-moment microphysics LIMA v1.0 scheme of Meso-NH, Geoscientific Model Development, 11, 4269–4289, https://doi.org/10.5194/gmd-11-4269-2018, 2018.*

*Korolev, A. and Leisner, T.: Review of experimental studies of secondary ice production, Atmospheric Chemistry and Physics, 20, 11 767–11 797, https://doi.org/10.5194/acp-20-11767-2020, 2020.*

*Lachapelle, M., Nichman, L., Girouard, M., Nguyen, C., Ranjbar, K., Bliankinshtein, N., Bala, K., Thériault, J.M., French, J. R., Minder, J. R., Wolde, M.: Airborne and ground measurements for vertical profiling of secondary ice production during ice pellet precipitation J Atmos Sci, 10.1175/jas-d-24-0117.1, 2025.*

*Phillips, V. T. J., Yano, J.-I., and Khain, A.: 1. Ice Multiplication by Breakup in Ice–Ice Collisions. Part I: Theoretical Formulation, J. Atmos. Sci., 74, 1705–1719, https://doi.org/10.1175/JAS-D-16-0224.1, 2017b.*

2. Line 390: The Flight 20 case is a little confusing. The authors mention that the system primarily consisted of stratiform clouds, but Fig. 12 suggests a deeper system with cloud top heights near 8-9 km above ground. Also, the method of ice formation is claimed to be primarily homogenous freezing and precipitation of ice particles from above. It isn't clear from Fig 12 radar data if such ice precipitation is observed. Could the authors provide more details regarding the claim of Ni primarily from precipitating ice vs heterogeneous freezing within the storm system?

*Thank you for this comment! After further analyzing the data, we changed our claim here.*

*Thank you for this comment! After further analyzing the data, we changed our claim here, and added discussions below in the manuscript:*

*"One possible reason for the fairly accurate $N_i$ values in the F20-BASE simulation is that in this stratiform case $N_i$ is likely governed by the heterogeneous nucleation process, with SIP playing a relatively less important role. In the F20-BASE simulation, most $N_i$ values between –15°C and –25°C (T of the maximal flight altitude for Flight 20 at ~6.4 km) are higher than the parameterized $N_i$ due to condensation freezing/deposition ice nucleation (lower half of the small red cycles in Figure 18a) within the same temperature range, but lower than the parameterized $N_i$ for colder temperatures (upper half of the small red cycles in Figure 18a). Ice particles formed at temperatures below –25°C through heterogenous nucleation may gradually fall to warmer, lower-altitude regions, thereby contributing to higher $N_i$ compared to the parameterized $N_i$. Additionally, simulated results from F20-BASE (Figure 18a) show a gradual decrease in $N_i$ with increasing temperature, underscoring the effect of ice particle aggregation.*

*In the case of Flight 20, the cloud top extends above the homogeneous freezing threshold at -40°C, allowing for the possibility that homogeneous ice nucleation could also influence $N_i$ at lower altitudes as these ice particles descend. However, as shown in Figures 18a, the $N_i$ values above the -40°C level are not substantially higher than most $N_i$ values seen between -20°C and -30°C (ranging from $10^3$ to $10^5$ m$^{-3}$). As ice particles formed through homogeneous freezing descend into warmer temperatures, aggregation processes are expected to further decrease Ni, resulting in concentrations lower than those shown between –20 °C and –30 °C in Figure 18a, where most values range from $10^3$ to $10^5$ m$^{-3}$. This pattern indicates that, at temperatures warmer than -25°C, the ice particle concentrations are more likely dominated by heterogeneous nucleation rather than by the effects of homogeneous nucleation.*

*Compared to F20-BASE (Figure 18a), F20-HMgc-FFDh (Figure 18b) and F20-HMgc-FFDl-CB (Figure 18c) have more frequent occurrences of higher $N_i$ values for temperatures above –10°C, indicating the influence of SIP. However, these impacts remain moderate.*

*In the observed $N_i$ frequency distribution (Figure 18d), two distinct clusters are apparent. One cluster (indicated by the large red circle) aligns closely with the parameterized $N_i$ line, suggesting the influence of heterogeneous ice nucleation. The second cluster, which exhibits higher $N_i$ values below –20°C, may indicate the presence of falling ice particles originating*

*from higher altitudes, regions that were not directly observed due to the flight's altitude limitations. The higher $N_i$ values may also result from the effects of SIP. However, $N_i$ gradually decreases with increasing temperature. Unlike the F20-HMgc-FFDh (Figure 18b) and F20-HMgc-FFDl-CB (Figure 18c) cases, there is no distinct 'protruding' cluster (green circles in Figure 18b, 18c) that would indicate a sudden $N_i$ increase at lower altitudes due to SIP. Thus, ice formation in warmer regions (lower altitudes) in Figure 18c is likely dominated by heterogeneous nucleation, although the influence of SIP cannot be ruled out.*

*Observed $N_i$ values at $T > -20°C$ (Figure 18c, $N_i > 1 \times 10^4$ $m^{-3}$) are frequently higher than those in the simulations (Figures 18a, 18b and 18c, where $N_i$ mostly ranges between $1 \times 10^3$ and $1 \times 10^4$ $m^{-3}$). This discrepancy warrants further investigation into the uncertainties associated with the parameterized heterogeneous nucleation at colder temperatures ($T < -20°C$), the ice–ice collection efficiency and SIP."*

[Figure]

*Figure 18: 2D histograms for the frequency of Ni regarding the T for F20-BASE (panel a), F20-HMgc-FFDh (panel b), F20-HMgc-FFDl-CB (panel c), and the observation for Flight 20 (panel d). Small red cycles: parameterized $N_i$ due to condensation freezing/deposition ice nucleation (adapted from Cooper, 1986). Red dashed line: minimal flight T between 12:30 and 16:00 UTC.*

3. Line 425: Can the authors expand on the reasoning why storm system longevity might contribute to HIWC? Does a growing ice particle feel the storm system lifetime? At first thought, it should fall out of the cloud whenever it gets too heavy, and stronger updrafts should help it stay afloat longer and get bigger. This suggests that stronger updrafts may result in larger ice particles, yet Fig 19 suggests that larger ice particles were observed during weaker updraft conditions. Some explanation/clarification will better help the readers since these are very interesting results.

*This is a great question! We have several hypotheses to explain the differences. These discussions are added into the manuscript:*

*"In the Cayenne case (tropical setting), the storm system exhibits much stronger convection, with more intense turbulence and updrafts, which in turn enhances SIP processes. For instance, vigorous updrafts and abundant supercooled liquid water near the melting layer increase the mixing of liquid droplets and ice particles, thereby boosting both HM and FFD rates. Turbulence further amplifies the efficiency of CB. Overall, the $N_i$ in the tropical case (Cayenne) is higher than in the mid-latitude winter cases (ICICLE F9 and F20), resulting in a size distribution that is more heavily weighted toward smaller particles for Cayenne.*

*Strong updrafts in tropical mesoscale systems limit the growth time of ice particles, as many are rapidly transported to higher altitudes. While these vigorous updrafts can promote the formation of rimed particles such as graupel and hail, such particles are generally confined to localized regions within the storm, and the majority of ice in tropical systems consists of smaller ice particles. In contrast, ice particles in mid-latitude winter nimbostratus clouds tend to have longer residence times, allowing for more favorable conditions for depositional growth and aggregation. As a result, aggregation processes are more dominant in mid-latitude winter storms, leading to a greater abundance of larger snow aggregates.*

*Note that although the fraction of smaller ice particles in the Cayenne case is higher than in ICICLE F20, the total IWC in the Cayenne case is actually greater, with some extreme values exceeding 3 g m$^{-3}$, levels not observed in ICICLE F20."*

4. Line 435: Could HIWC be influenced by higher and lower aerosol number concentrations during the French Guiana, ICICLE F09 and F20 flights? Could aerosol cleansing during a longer storm increase supersaturation fluctuations to form bigger ice particles during F20?

*In a recent experiment using GEM for the French Guiana case, we reduced the background aerosol number concentration from the default value in the P3 scheme (300 cm$^{-3}$) to 80 cm$^{-3}$. This adjustment led to a decrease in liquid drop number concentration, and although the LWC also decreased, it did so less significantly, resulting in generally larger liquid drops. Because the FFD process accelerates with increasing drop size (being proportional to $D^4$), this produced higher $N_i$ values. However, the impact on IWC was mixed: a decrease was observed below 8 km, while an increase occurred above 8 km. These results suggest that aerosols do influence cloud microphysical processes, but further studies are needed to fully understand their potential impacts.*

**Minor corrections:**

Line 113: How thick is the observed melting layer for F09 and F20? What is the model vertical resolution and is it able to resolve the melting layer for each grid size?

*Radar observations indicate that the melting layers for both flights occur at roughly the same altitude, around 2 km, with a thickness between approximately 200 and 500 m. The model's vertical resolution at this altitude is about 250 m, which should be sufficient to resolve the melting layer. In Fig. 17 (or Fig. 18 in the revised version), the enhanced reflectivity associated with the melting layer is clearly identifiable near 2 km altitude.*

Line 149: Is this a typo? The HM peak is at -5 C, and also goes to 0 at -5 C, Should it be -8 C?

*Sorry about the confusion. This is a typo indeed. We changed the text into:*

*"The parameterized HM process follows Hallet and Mossop (1974) stating the production of a peak value of 350 ice splinter per mg of collected liquid water during riming of liquid cloud drops or rains droplets within a temperature range of -3°C > T > -8°C, with the peak value at -5°C and varying linearly to 0 at -3°C and  **-8** °C."*

Line 164: What is numerical value with units of the max Nf rate at T = -12.5 C?

*The maximum rate of $N_f$ at T=-12.5°C is the value directly calculated by Eq. 3 which depends on the size of the liquid drop (D). This $N_f$ will then be linearly scaled down to zero at T=-25°C and T=-2°C. We updated the description:*

*"Following Keinert et al. (2020) the activity of the FFD process was limited to the temperature range -25°C < T < -2°C with a maximal rate of $N_f$ (as calculated by Eq. 3, with or without capping) at T = -12.5ºC. $N_f$ is then linearly scaled down to zero as the temperature decreases from −12.5°C to −25°C or increases from −12.5°C to −2°C, according to a T dependent scaling factor c (where $c \in [0,1]$), such that the rate is $c \times N_f$. All newly produced ice splinters have a diameter of 10 µm."*

Line 173: The "which collect rain droplets" makes the sentence a little difficult to read.

*Thanks for this suggestion. We would like to retain this specification, as omitting "that collected the rain" could be misleading. There are multiple ice categories, and not all are involved in rain collection as referenced here.*

*"Excluding the mass of ice splinters, which typically represents a small fraction of the rain mass, the remaining rain mass is added to the ice category that collected the rain."*

Fig 2: Can the F09 track also be overlayed on this figure? It will help the reader locate flight observations, the storm system and model grid centers better.

*Thanks for this suggestion! Please find below the updated Fig. 2 and Fig. 11, with the flight tracks shown as green lines.*

[Figure]

*Figure 2. (a) GOES-16 spectral radiance at 0.64 μm (Channel 2). (b) GOES-16 brightness temperature at 10.3 μm (Channel 13). Both images are from 20:30 UTC on 9 February 2019. Blue lines denote coastlines and lakes; the red rectangle indicates the inner-most simulation domain (with a 0.25 km grid spacing) for Flight 9; yellow lines show the sea level pressure isobar at 18:00 UTC from the operational regional analysis used at ECCC; green lines: aircraft flight track. (Only data from the flight segments in the northern portion of the track were used in this study).*

[Figure]

*Figure 12. (a) GOES-16 spectral radiance at 0.64 μm (channel 2). (b) GOES-16 brightness temperature at 10.3 μm (channel 13). Both images are from 14:30 UTC on 23 February 2019. Blue lines represent coastlines, green lines represent aircraft flight track, and the red rectangle represents the inner-most simulation domain at 0.25 km resolution for Flight 20.*

Flight 224: "The simulated brightness temperature closely matches the GOES-16 observations." With the naked eye, the GOES-16 temperature looks lower for the cloud system. Could you provide mean temp values for both for the cloud system?

> *Figure 4 is intended to illustrate how GEM simulates sea-level pressure and cloud cover at the synoptic scale. The model outputs shown here are from a simulation at lower horizontal resolution (2.5 km), as the 0.25 km resolution domain is too limited for this scale of comparison. As you also observed, there is a brightness temperature bias in the simulated results when compared to GOES-16 values. Investigations into the cause of this bias are*

*ongoing. However, presenting the brightness temperature bias at 2.5 km resolution may be beyond the scope of this paper. We have added a clarification in the manuscript:*

*"The simulated brightness temperature at 10.3 μm closely matches the GOES-16 observations. However, a positive bias (indicating warmer values) is evident compared to the satellite data. Investigations into the cause of this bias are currently underway. Additionally, the simulation slightly underestimates cloud coverage over the southern part of Lake Michigan and in the southeastern corner of the simulation domain."*

Line 304: It isn't clear exactly where the enhancement is occurring in the figure. Could you add a box/arrow to point at the regions?

*The enhancement region is indicated by the white circles in the updated Fig. 10 (previously Fig. 9) (now the results of HMgc-FFDl-CB are also included):*

[Figure]

*Figure 10. Averaged values between altitude of 0.5 and 3.8 km for Flight 9. (a), (b), (c): IWC; (d), (e), (f): Ni. First column for F9-BASE, middle column for F9-HMgc-FFD, third column for F9-HMgc-FFDl-CB. Red line: flight track.*

Line 315: Also unclear where the increase in IWC is to be seen. The scale is massive and the difference in shades of grey is tough to spot.

*The enhancement region is highlighted by the white circles in the updated Fig. 11 (previously Fig. 10) (now the results of HMgc-FFDl-CB are also included):*

[Figure]

*Figure 11. RWC (a, b, c) and IWC (d, e, f) near the surface for F9-BASE (a, d), F9-HMgc-FFDh (b, e) and F9-HMgc-FFDl-CB (c, f) simulations at 20:30 UTC on 2019-02-07. Black contour: region of freezing rain (RWC > 0.01 g m-3 and T < 0°C in the lowest model layer at ~20 m above surface). Red lines: Flight 9 aircraft track. Yellow lines: part of flight track below 800 m of altitude observing both supercooled liquid and ice near 20:12 UTC. Turquoise lines: part of flight track below 800 m of altitude observing only ice near 20:47 UTC.*

---

## Author Response (AR2)

**Answers to Referee #1: Xiaohong Liu, xiaohong.liu@tamu.edu**

Thank you for the additional comments and the helps with improving the quality of this manuscript! Please find the responses below to each of the comments.

Referee comments:

I appreciate the authors to make good efforts to address my comments on the manuscript, particularly improving the introduction to the model parameterizations, and quantifying the uncertainty of the FFD parameterization, and testing the third SIP process (ice-ice collisional breakup (CB)). While the manuscript is in a good shape for acceptance, I still have some minor comments/corrects before the publication.

1. Line 94, "particles sizes" -> "particle sizes".

Thank you! Corrected.

2. Line 142. "Because of this, each ice category in P3 can represent any type of ice-phase hydrometeor (with the limitations of a bulk scheme)." This sentence is a bit confusing. Can you be clear here what the limitations a bulk scheme are?

Thank you for the comment, we adjusted the phrase as:

"Because of this, each ice category in P3 can represent any type of ice-phase hydrometeor, although, as in any bulk scheme, ice categories in P3 remain constrained by assumed PSD parameters type and the number of prognostic moments."

3. Line 158-159. "..., immersion/contact freezing of cloud droplets and rain drops (see Morrison and Milbrandt, 2015), ..." Can you give some detail of treatment of immersion/contact freezing based on Morrison and Milbrandt, 2015? Is it dependent on aerosol, or only on temperature?

We modified the phrase as:

"..., immersion/contact freezing of cloud droplets and rain drops (temperature-dependent following Bigg (1953) with parameters from Barklie and Gokhale (1959) ), ..."

Two references are added:

Barklie, R. H. D., and Gokhale, N. R.: The freezing of supercooled water drops. Alberta hail, 1958, and related studies, McGill University Stormy Weather Group Sci. Rep. MW-30, Part III, 43–64, 1959.

Bigg, E. K.: The supercooling of water. Proc. Phys. Soc., 66B, 688–694, doi:10.1088/0370-1301/66/8/309, 1953.

4. Line 159. "heterogeneous nucleation via condensation freezing and deposition (Cooper, 1986)," please rewrite this to something like: condensation freezing of cloud droplets and rain drops (?) and deposition nucleation.

We changed the phrase to: "Heterogeneous nucleation through condensation freezing of cloud droplets and rain drops, and deposition nucleation (Cooper, 1986)."

5. Figure 3. As I mentioned in the first round of my review, the unit (Y-axis) of Ni seems not correct. Currently it reads as 0-5 m^-3, while it should be 10^0 - 10^5 m^-3, if I understand correctly. This also applies to other similar figures.

Sorry that we missed this point! Thank you again for mentioning it! Fig. 3 and 13 are now updated.

6. Line 486-517. You talk about the heterogeneous nucleation (condensation freezing/deposition ice nucleation) for the importance of comparing the modeled and observed T - Ni histogram (Figure 18) in this flight and also include the red open circles from Cooper 1986. How about the role of immersion/contact freezing of cloud droplets and rain drops? In mixed-phase clouds, immersion/contact freezing could dominate the ice nucleation based on previous studies.

We agree that immersion and contact freezing are another source which could explain the higher Ni values compared to the Cooper (1986) line (small red dots). We adjusted the paragraphs as:

"One possible reason for the fairly accurate Ni values in the F20-BASE simulation is that in this stratiform case Ni is likely governed by the heterogeneous nucleation process, with SIP playing a relatively less important role. In the F20-BASE simulation, most Ni values between –15°C and –25°C (T of the maximal flight altitude for Flight 20 at ~6.4 km) are higher than the parameterized Ni due to condensation freezing/deposition ice nucleation (lower half of the small red cycles in Figure 18a) within the same temperature range, but lower than the parameterized Ni for colder temperatures (upper half of the small red cycles in Figure 18a). Ice particles formed at temperatures below –25°C through condensation freezing/deposition ice nucleation may gradually fall to warmer, lower-altitude regions, thereby contributing to higher Ni compared to the parameterized Ni. Another possible source of the enhanced Ni is immersion and contact freezing. Additionally, simulated results from F20-BASE (Figure 18a) show a gradual decrease in Ni with increasing temperature, underscoring the effect of ice particle aggregation."

"In the observed Ni frequency distribution (Figure 18d), two distinct clusters are apparent. One cluster (indicated by the large red circle) aligns closely with the parameterized Ni line, suggesting the influence of condensation freezing/deposition ice nucleation. The second cluster, which exhibits higher Ni values below –20°C, may indicate the influence of immersion and contact freezing, as well as the presence of falling ice particles originating from higher altitudes, regions that were not directly observed due to the flight's altitude limitations. The higher Ni values may also result from the effects of SIP. However, Ni gradually decreases with increasing temperature. Unlike the F20-HMgc-FFDh (Figure 18b) and F20-HMgc-FFDl-CB (Figure 18c) cases, there is no distinct 'protruding' cluster (green circles in Figure 18b, 18c) that would indicate a sudden Ni increase at lower altitudes due to SIP. Thus, ice formation in warmer regions (lower altitudes) in Figure 18c is likely dominated by heterogeneous nucleation, although the influence of SIP cannot be ruled out."